# Investigation of coastal sea-fog formation using the WIBS (Wideband Integrated Bioaerosol Sensor) technique

5   Shane M. Daly[1], David J. O'Connor[2], David A. Healy[1], Stig Hellebust[1], Jovanna Arndt[1], Eoin J. McGillicuddy[1], Patrick Feeney[1], Michael Quirke[1] John C. Wenger[1], John R. Sodeau[1]

10  1. School of Chemistry and Environmental Research Institute, University College Cork, Ireland.

    2. School of Chemical and Pharmaceutical Sciences, Dublin Institute of Technology, Dublin, Ireland.

    Correspondence to: John R. Sodeau (j.sodeau@ucc.ie)

Keywords: Iodine, fluorescence detection, optical microscopy, marine aerosols

35

***Abstract.***

A Wideband Integrated Bioaerosol Sensor (WIBS-4) was deployed in Haulbowline Island,
Cork Harbour to detect fluorescence particles in real-time during July and September,
2011. A Scanning Mobility Particle Sizer (SMPS) was also installed providing sizing
analysis of the particles over the 10 - 450 nm range. During the campaign, multiple fog
formation events occurred; they coincided with dramatic increases in the recorded
fluorescent particle counts. The WIBS sizing/fluorescence intensity profiles indicated that
the origin of the signals was potentially non-biological in nature (*i.e.* PBAP, Primary
Biological Aerosol Particles). Furthermore, the data did not support the presence of known
fluorescing chemical particles like SOA (Secondary Organic Aerosol). Complementary
laboratory studies showed that the field results could potentially be explained by the
adsorption of molecular iodine onto water droplets to form $I_2 \cdot (H_2O)_x$ complexes. The
release of iodine into the coastal atmosphere from exposed kelp at low-tides has been
known for many years. This process leads to the production of small $I_xO_y$ particles which
can act as Cloud Condensation Nuclei (CCN). While the process of molecular iodine
release from coastal kelp sources, subsequent particle formation and the observations of
sea-mists and -fogs have been studied in detail, this study provides a potential link between
the three phenomena. Of mechanistic interest is the fact that molecular iodine included into
(rather than on) water droplets does not appear to fluoresce as measured using WIBS
instrumentation. The study indicates a previously unsuspected stabilizing transport
mechanism for iodine in the marine environment. Hence the stabilization of the molecular
form would allow its more extensive distribution throughout the troposphere before
eventual photolysis.

**Introduction**

Atmospheric aerosols comprise a variety of chemical and biological components released from natural and anthropogenic sources. The complexity of the resulting dispersions is a direct result of their individual compositions and the varied processes that control their physical forms and atmospheric transformations. Their effects on our health, ecology and climate are now well documented although our capabilities for predicting their exact roles in such phenomena are still not well-understood. One of the most significant sources of atmospheric aerosol is the oceans because they cover >70% of the Earth's surface and carry >90% of all its saline water. Most importantly they are locked into a continuous, dynamic interplay with Earth's land and atmosphere.

Marine aerosols do not consist of sea-salt alone. They contain organic content from fatty alcohols and acids, sulfur compounds originating from phytoplankton and also varying quantities of magnesium and calcium (Fitzgerald, J. W., 1991, Gagosian et al., 1986, Savoie, D. L. et al., 1980). They are released into the air by bubble-bursting processes which produce sea-spray droplets that evaporate leading to a wide distribution of particle sizes ranging from ~10 nm to many microns. These dispersions also represent some of the first aerosol types to be systematically investigated because of the historical observations of sea-fogs. Their formation was eventually ascribed to the involvement of cloud condensation nuclei (CCN), which promoted the formation of cloud droplets in air at humidities close to 100% (Twomey et al., 1959., 1967).

Relatively transparent coastal mists have also long been noted and observed to act as precursors to the development of fogs that form at a humidity as low as 70% (Twomey et al., 1955). These coastal mists represent a situation when condensation and evaporation are competing in a region where breaking waves are dominant thereby leading to the release of large densities of sea-salt particles. However, it is now well-established that the CCN underpinning the coastal mists and fogs are not all released by this relatively simple bubble-bursting mechanism. Another source driving the processes is the emission of iodine from kelp.

Since the discovery of iodine by Courtois in 1811, who isolated it from kelp ash, the element and its compounds have been shown to play an extensive role in our lives: from

the maintenance of good health to the control of atmospheric chemistry and composition. In regard to the marine aerosol, there are various species of seaweed that are able to emit iodine and iodocarbons to the air when they become exposed to the atmosphere. They include two main emitting sea kelp types, *Laminaria digitata* and *Laminaria hyperborean*, with *Laminaria digitata* being the most potent emitter of iodine (Ball et al., 2009; Huang et al., 2010; Monahan et al., 2012). When the sea kelp is exposed, it undergoes oxidative stress from low level traces of ozone in the atmosphere (Palmer et al., 2005; Küpper et al., 2008; Monahan et al., 2012). Solar radiation can also stress the kelps in the daytime when tidal levels are observed to be at their minimum. (Seitz et al., 2010).

The initial, air-oxidised products include HOI, IO and OIO, which can all interact and agglomerate to yield a variety of $I_xO_y$ polymer particulates with dimensions in the nano/sub-micron region (Saiz-Lopez et al., 2011). Therefore, under the correct circumstances, they are able to act as CCN capable of initiating a coastal mist (Burkholder J. B. et al., 2004; Hoffmann et al., 2001; Küpper et al., 2008; O'Dowd et al., 2002).

The most important condition for iodine oxide particulate releases, other than the presence of appropriate kelp types, is a low sea-tide level because they play an important role in exposing the kelps to the atmosphere. The occurrence of Neap and Spring Low Tides are crucial in this respect because they can lead to total exposure of the kelps. Spring tides are especially effective in promoting their exposure, including the lower lying, iodine-releasing *Laminaria* species (*digitata* and *hypoborean*). However, the majority of tides only reveal the mid-lying sea kelp species e.g. *Ascophyllum nodosum* and *Fucus vesiculosus*.

Real-time measurements of the number-concentrations and types of airborne primary biological aerosol particles (PBAP) released in contrasting world-wide locations including urban, tropical and rural environments have dramatically increased in number over the last ten years. The main reason for the scientific attention is the relatively recent development of instrumentation capable of counting individual bioaerosol particles in the millisecond timescale (Fennelly et al., 2018). The technique is based on the detection of biofluorophores, such as amino acids, tryptophan and NAD(P)H, that are present in bacteria, pollen and fungal spores (O'Connor et, al 2014., O'Connor et, al 2011). Light scattering is also employed to probe the individual particles and thereby give size

distributions and some indication of shape. The differing discriminatory measurements give a good estimate of the proportion of biological particles in a full ensemble that always consists of many chemical, non-fluorescent solids. The technique has been utilized in a number of region and contacting environments (Feeney et al., 2018., Fernández-Rodríguez et, al 2018, Perring et al 2014., Gabey et al., 2010). However, it should be noted that some chemical solids do fluoresce such as Secondary Organic Aerosol (SOA) but those characterized to date are generally very small (< 1 μm) and are mainly present in locations where there are large anthropogenic sources (Pöhlker et al., 2012., Toprak et al., 2013). To further aid discrimination, a variety of data-filtering methods to count the particles most likely to be biological in nature have recently become established (Hernandez et al., 2016). In the current study the forced trigger (FT) + 3 σ method was applied as a filter. Although other methods of data analysis have begun to used for such analysis (Savage et, al 2018).

The results of the campaign reported here are related to a study mounted to obtain real-time number-concentrations of PBAP released at a coastal site *i.e.* a location never investigated before in this context.

## 2 Methodology

### 2.1 Campaign Site

A sampling campaign appropriate to measure number-concentrations of PBAP in real-time was mounted at Haulbowline Island, situated within Cork harbour (51.8406° N, 8.3028° W), an active Naval Base distanced ~ 6 km from the Irish Sea. The on-line particle detection instrumentation comprised a Wideband Integrated Bioaerosol Spectrometer (WIBS-4) and a Scanning Mobility Particle Sizer (SMPS). Both were housed in a trailer situated in the NNW corner of the island and 5-10 m from steps leading down to the shoreline. These coastal access steps were covered with various sea weed/kelp types. Mid-lying tides revealed *Fucus vesiculosus* and *Ascophyllum nodosum* on the higher steps (~2 m above sea level). *Laminaria digitata* and *Laminaria hyperborean* were found on the lowest steps (~1 m above sea level) becoming clearly visible during the low tides of July, which ranged between 0.46 m on 16th July, 0.7 m on 26th July and 1.3 m on 23rd July. (Nash et al., 2008). Air monitoring was performed for a six-week period during the summer of

2011 with the specific dates analysed in detail for this study being 15/16 July, 23/24 July and 26/27 July 2011.

The island itself is located 1-2 km due south from Cobh town which is a popular stop-off point for cruising ship carriers. An active crematorium resides on another island 1-2 km due south of the sampling site. Due to its military function vegetation is scarce on Haulbowline Island although meteorological information from Met Éireann, including visibility measurements, are readily available for the locality.

*2.2. Field Instrumentation*

The Wideband Integrated Bioaerosol Sensor model 4 (WIBS-4) is a prototype real-time biological particle sensor developed by the University of Hertfordshire in the UK and used in many field and laboratory campaigns directed toward the detection of bioaerosols over the last ten years. (Kaye et al., 2005; Gabey et al., 2010; Stanley et al., 2011, Healy et al., 2012a, b). Its mode of operation, strengths and weaknesses have been described in many publications and only a brief summary of its main features is given here. Air is pumped into a central optical chamber at a rate of 2.4 L/min where a continuous-wave 635 nm diode laser is used for the initial detection of the particles and their sizing. An Asymmetry Factor (AF) giving an indication of the sphericity of the individual particles is also obtained at this point. Then two pulsed, filtered xenon flash lamp UV sources (280 nm and 370 nm) are fired sequentially. Any fluorescence emission resulting from excitation of a particle is then collected in two wavelength bands: 310-400 nm and 420-650 nm. As a result, data for three fluorescence channels are obtained: (i) excitation at 280 nm, emission 310-400 nm (FL1 Channel); (ii) excitation at 280 nm, emission 420-650 nm (FL2 Channel); (iii) excitation at 370 nm, emission 420-650 nm (FL3 Channel). The size range that could be detected with the WIBS-4 used in this study was between 0.5 - 22 µm. Fluorescence thresholds were determined by placing the instrument into forced trigger mode. The mean fluorescence intensity plus three times the standard deviation of the collected values whilst in this mode of operation were utilized as a threshold in each of the channels (FL1,2 and 3). A WIBS-4A (Droplet Measurement Technologies) was utilised in the laboratory chamber studies. Both instruments display similarities in terms of sampling methods and build but have a few distinctions such as the WIBS-4 dual gain detection approach and the WIBS-4A double threshold system. The WIBS-4A has a slightly higher flow rate at 2.5 L/min and ~300 ml/min (flow velocity of ~18 m/s) compared to the WIBS-4 at 2.4 L/min

and ~230 ml/min (flow velocity of ~12 m/s). Similarly, variation in fluorescent intensity between WIBS instruments for identical particles is a potential problem in such studies a problem which has been discussed previously in the literature. (Robinson et al., 2017, Savage et al., 2017, and Könemann et al., 2018)

A Scanning Mobility Particle Sizer (SMPS, TSI Series 3080), was also deployed to give a profile of particles present that would be too small to be detected by the WIBS-4. The instrument is capable of sizing particles in the 10 - 478 nm range. Its operation has been described in many previous reports such as for the generation of mono-disperse flows by classification of incoming particles using a differential mobility analyser (DMA) and a condensation particle counter (CPC). (Kidd et al., 2014; Mills et al., 2013)

Visibility measurements were taken from the Met Éireann monitoring facility at Roches' Point ~6 km due south from the campaign site in order to record information about any local mist/sea-fog formation. These data are obtained by use of a continuous laser fired at a range of 50,000 m across land. Essentially, in clear skies no scatter signal can be recorded but in the event of a fog the condensed water molecules present cause quantifiable scattering of the laser light.

Data analyses were performed using a number of different computational programmes. A combination of Excel and IGOR provided graphical plots of data recorded by the spectroscopic instrumentation. The AIM (Aerosol Instrument manager) software programme was used to run and handle data from the SMPS. An in-house designed Matlab toolkit was used for statistical analysis of the very large datasets obtained over the campaign. Using this approach trends in particle fluorescence and size evolution were immediately realisable.

*2.3. Laboratory experiments*

A set of laboratory mimic experiments were performed in order to reproduce the results obtained in the field campaign. Hence, a UV-Vis Spectrometer (Thermo Fisher Scientific Model EVO 60) was used to determine the spectra resulting from a number of iodine-containing solutions in the absence and presence of chloride ions. A set number of mixtures were prepared using combinations of three main components; iodine (pellet form from Sigma-Aldrich, ≥99% purity), water (Milli Q) and sea salt, in order to mimic the main

components in sea-spray. 0.25g of ≥99% iodine crystals were measured for each sublimation test, with 0.05 g of refined rock salt from the Wieliczka Salt Mine used for the saltwater mimic tests. A much smaller quantity of salt was used to avoid overloading the detector.

The following composition was present in 35 ppt of saltwater solution (3.5 g of rock salt in 100 ml of water): $Cl^-$: 55.29%, $Na^+$: 30.74%, $Mg^{2+}$: 3.69%, $SO_4^{2-}$: 7.75%, $Ca^{2+}$: 1.18% and $K^+$: 1.14%. In all, the $Cl^-$ and $Na^+$ ions make up 86% of the total salinity.

10 Fluorescence spectra of the solutions were investigated using a Shimadzu RF-6000. The nebulized counterparts were analysed using the WIBS-4 coupled to a reaction chamber in an arrangement based on one previously used to successfully investigate pollen and fungal spore dispersions (Healy et al., 2012a; 2012b).

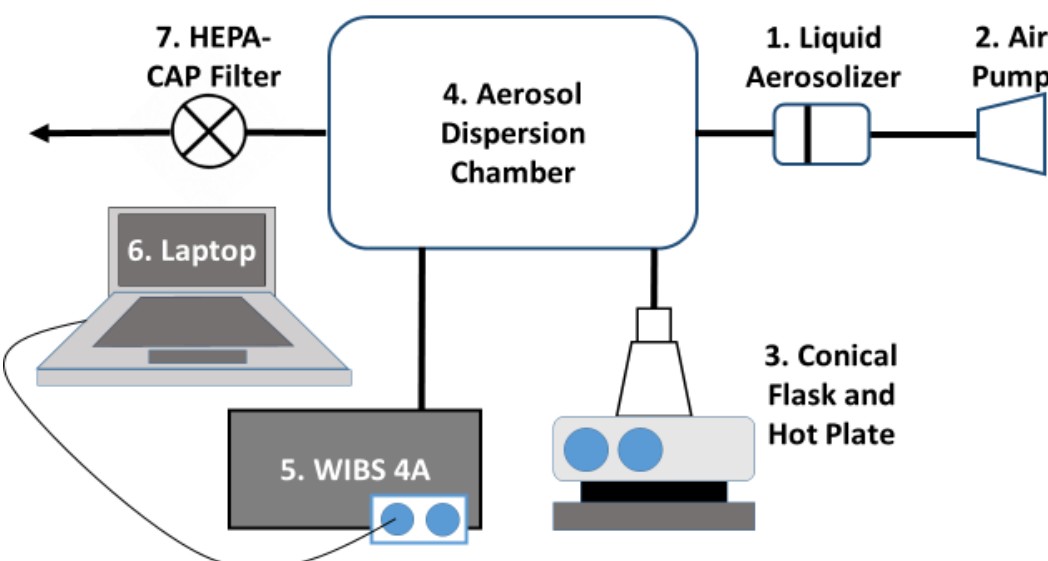

*Figure 1.* **Schematic diagram of the chamber set-up for the aerosolization experiments.**

20 The schematic of the chamber set-up is shown in Figure 1. It consisted of a liquid aerosolizer to pump-disperse the Milli-Q water and salt/water droplets into the 15 L dispersion chamber to which iodine vapour could be introduced after an appropriate mixing time by sublimation of set amounts of iodine crystals. The type of nebulizer used could

have a large effect on the water droplet size but this would not be expected to interfere with the detection of fluorescence signals.

The chamber was rigorously cleaned between experiments to avoid sequential iodine staining and contamination of internal WIBS surfaces. Before each experiment, the system was pumped down for 30 minutes to remove any residual material using the WIBS-4A total pumping capacity of 2.5 L/min. During the experiment, a flow of ~5.6 L/min of compressed air was supplied for the aerosolization into the system. No pressure transducer was present in the system at the time so estimated conditions were of the order of ~1 atm at 298 K. Relative humidity was >70 % based from observation of the chamber rather than direct measurement. Separate experiments were performed under the same conditions using either pure iodine vapour or water droplets alone to act as controls. This study could be later applied to a flow tube experiment where there is a greater control of experimental conditions such as residence time.

The WIBS fluorescence data obtained in the experiments were filtered using thresholds most commonly utilized in the literature (i.e. the mean of forced trigger mode values + 3σ method) (Hernandez et al., 2016):

A: particle fluorescence with values above the fluorescence threshold for FL1 and below the fluorescence threshold in FL2 and FL3.

B: Greater than the FL2 fluorescence threshold but less than threshold for FL1 and FL3

C: Above FL3 threshold but less than FL1 and FL2 threshold.

AB: Dual filter: Above FL1 and FL2 threshold but less than FL3 threshold.

AC: Above FL1 and FL3 threshold but less than FL2 threshold.

BC: Above FL2 and FL3 threshold but less than FL1 threshold.

ABC: Above threshold for all three channels.

For the WIBS-4A instrument, size calibrations were carried out using Polystyrene Latex Spheres (PSL) with diameters 0.5, 0.82, 1, 2, 4, 10, 12 µm. The internal photomultipliers for each WIBS were not measured at the time.

## 3. Results and Discussion

*3.1 Haulbowline campaign*

On-line air sampling was performed at Haulbowline Island using the WIBS-4 and SMPS between 15-31 July and 1-30 September, 2011. A total number of 55,100,204 particles were recorded by the WIBS-4 with 663,835 of them exceeding the forced trigger threshold. Hence 1.2% of the total count were deemed to be fluorescent particles. By comparison, in most urban/semi-rural campaigns reported to date that value is in the range of 1-10%. (Huffman et al., 2010; Toprak et al., 2013; O'Connor et al., 2015). The fluorescent counts above threshold obtained in July were nearly double the amount of measurable counts in September although the overall counts were similar. Many factors may help to explain this finding, such as the increase in daytime hours between summer and autumn or seasonal differences in plant/algal/seaweed growth. The first period alone is analysed in detail in this report, primarily the July 15th-16th; 23rd-25th; 26th-27th dates as designated in Figure 2.

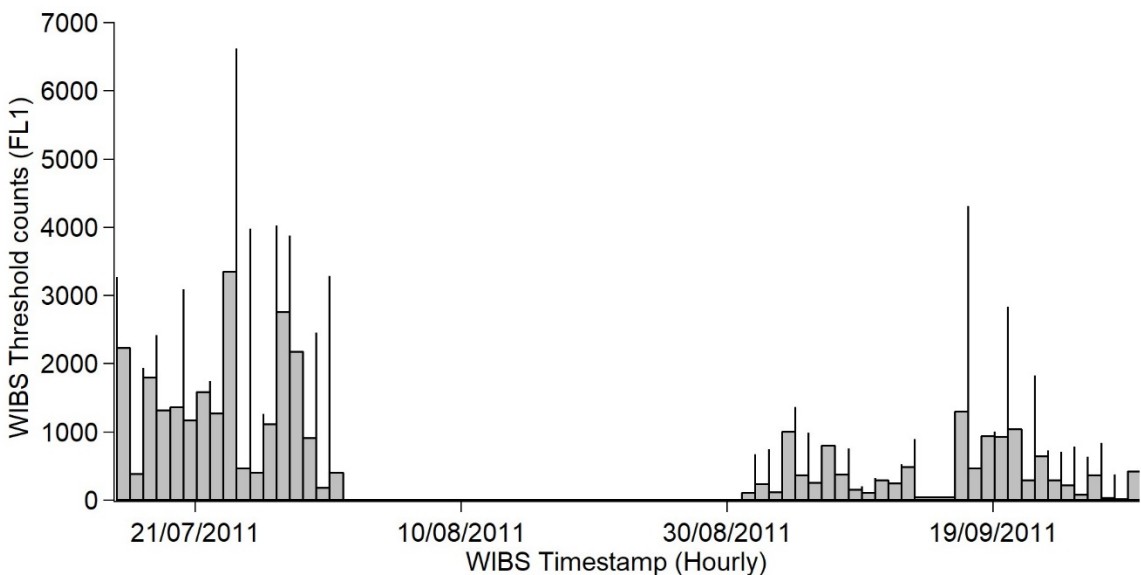

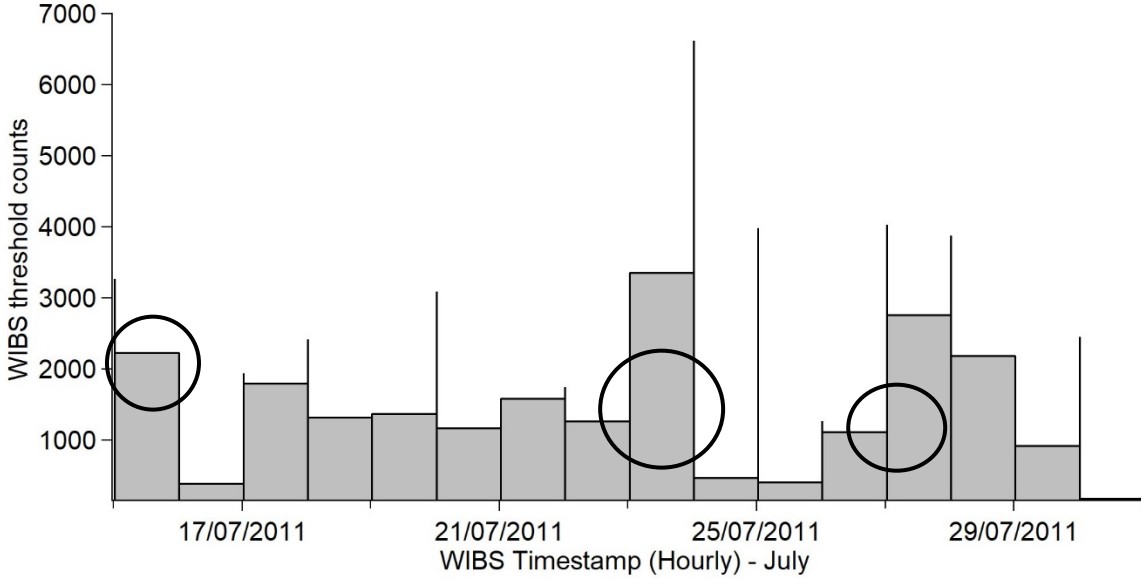

*Figure 2:* **Total campaign profile recorded by the WIBS (A) and the three events of July 2011 (B) discussed in this paper are highlighted with the black circles.**

The following baseline values were measured and used for the data analysis, FL1: 134, FL2: 20, FL3: 25. Analysis of the associated fluorescence/sizing data gave results that did not resemble any prior WIBS field campaigns. The profile is shown in Figure 3 for the FL1 channel data.

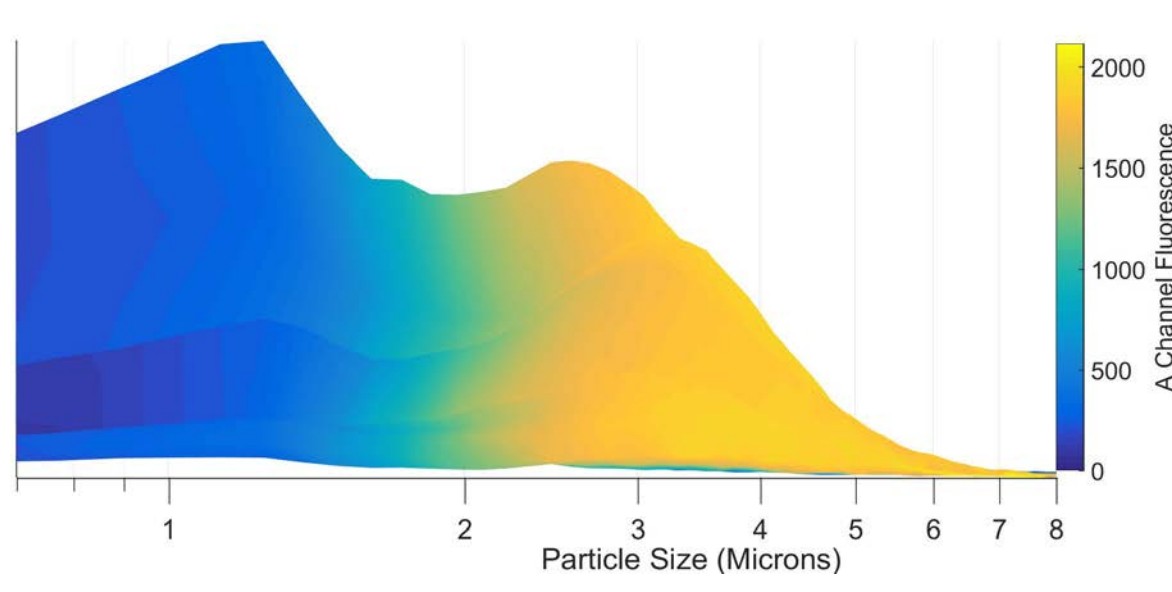

*Figure 3.* **Data visualization of FL1 fluorescence particle counts (after A channel filtering) recorded as a function of size during 15-16 July.**

It is clear, from the data shown in Figure 3, that a bimodal size distribution was recorded with: (i) a highly fluorescent, broad feature observed between ~2-6 µm, peaking at ~2.5 µm; (ii) a much narrower peak in the size regime <2 µm that represents 'weakly' fluorescent particles. From previous reports on coastal marine particles the smaller sized set can be identified, at least in part, as sea-salt solids or droplets. (O'Dowd et al., 1993). Sea-salt particles are known to quench other fluorescent species (Chmyrov A et al., 2010) so fluorescent particles in the size regime <2 µm are more difficult to observe. Fluorescence signals were mainly measurable in the FL1 channel. FL2 registered little emission above threshold as illustrated in Figure 4, which shows plots of size/AF data as a function of the FL1 and FL2 channels. (FL3 showed no fluorescence). The work by Hernandez et al (2016) suggests that some fungal spores show fluorescent characteristics that are present in FL1 but not FL3. However, the conditions on site would not favour spore release as the island has very little soil-based vegetation with only sea kelp present. Very low wind speeds were recorded during the measurement periods (<2.5 m/s during the 15[th] – 16[th] July and <5 m/s during the 26[th]-28[th] July). Therefore, it was highly unlikely that material could be carried on to the island from the mainland. In any case, the particles are less likely to be bacteria or pollen because of size constraints. Hence bacteria sizes are found at the lower limit (and below) of WIBS detection. Pollen sizes are often greater than the upper limit of WIBS detection and so are generally captured by the particle trap. During the summer period, the dominant fungal spore in locations close to but not at the Cork Harbour coastline, is known to be *Cladosporium* which is generally released during the day time (10:00 am - 12:00 pm onwards) and under dry conditions (O'Connor et al., 2015, Healy et al., 2014). By contrast in this study WIBS particle detection was found in the night-time period between 00:00 – 08:00 am.

Ascospores are linked with rain releases but only 0.2 mm of rainfall was recorded after 09:00 am on the 16[th] July, after which the WIBS signal is seen to decrease (O'Connor et al., 2015).

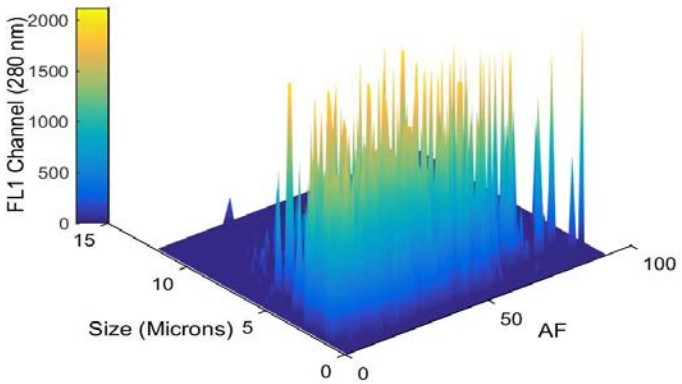

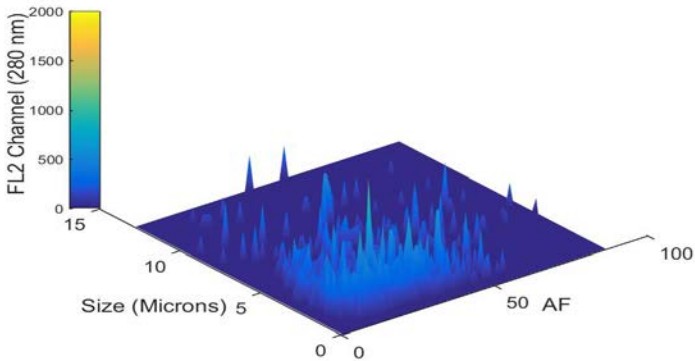

*Figure 4.* **Data visualizations of the FL1 and FL2 channel signals as a function of size and AF value during 15-16 July. The colour bar indicates fluorescent intensity with**
5 **the height of the peaks indicating particle counts.**

The larger size feature (2-6 μm) consisting of highly fluorescent solid particles/droplets but only in the FL1 channel represents a behaviour that has not been observed previously in any WIBS field campaign. Hence fungal spores, certain pollen and bacteria as large as
2 μm (Hernandez et al., 2016) can be found in the 2-6 μm size regime but are fluorescent in all channels because of their amino acid, tryptophan and NAD(P)H contents. Taken together these observations indicate that the fluorescent particles detected at Haulbowline Island were neither PBAP nor any other type of biological material. Furthermore, the relatively large size regime measured for the fluorescent particles coupled with the non-
fluorescent nature of the smaller particles would also appear to rule out anthropogenic influences from the port or crematorium.

The most intense fluorescent particle event was noted on 23-25 July 2011, when about double the numbers were observed compared to the 15-16 July measurements. The time evolution of the event is shown in Figure 5, where the 2-6 μm fluorescent particles grow between 03.00 and 08.00 on 24 July. Growth and decay behaviour is often seen in WIBS campaigns but no FL3 fluorescence was detected here, an observation which is clearly indicative of a non-biological origin for the particles. The distribution then declined to ~50% of its maximum number count over the four hours and the fluorescence was effectively extinguished by ~14.00. These data also show that the non-fluorescent particles below threshold (blue line), <2 μm, were less numerous overall than the fluorescent ones (red line) over this period and that particles of sizes > 6 μm were not present at all.

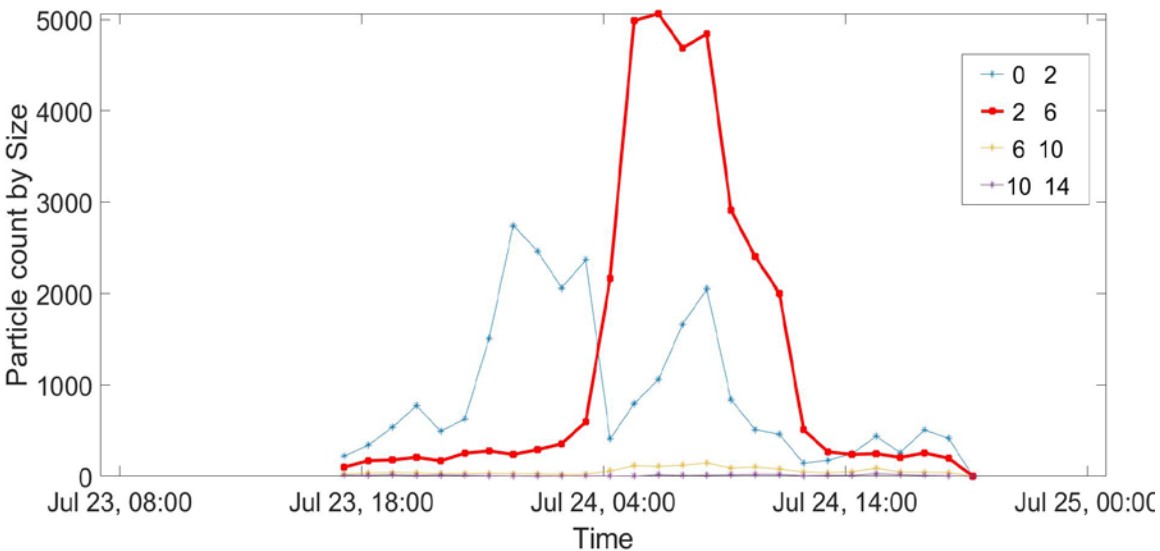

*Figure 5.* **Time evolution profile for the fluorescent (2-6 μm) and weakly-fluorescent particles (<2 μm) measured by WIBS between 23 and 25 July 2011.**

The behaviour of the non-fluorescent particles with dimensions < 2 μm contrasted strongly with that of the larger fluorescent particles. Hence between 23.00 on 23 July and 03.00 on 24 July there were few if any fluorescent particles but the non-fluorescent ones reached maximum counts. The rapid increase of the fluorescence signal at 03.00 coincided with a rapid decrease in the number of non-fluorescent particles for ~2 hours. These only started to grow again at ~08.00 when the fluorescent particles began their rapid decrease in numbers. After 14.00 few particles of either type (non-fluorescent <2 μm and fluorescent particles from 2-6 μm) were measured.

An SMPS data set, from 10 – 478 nm (0.01 – 0.48 μm), was also collected during 24-25 July 2011 in order to probe the behaviour of the smaller particles in the <0.5 μm size regime. This period represents a time period when a large fluorescence event occurred. The results are shown in Figure 6.

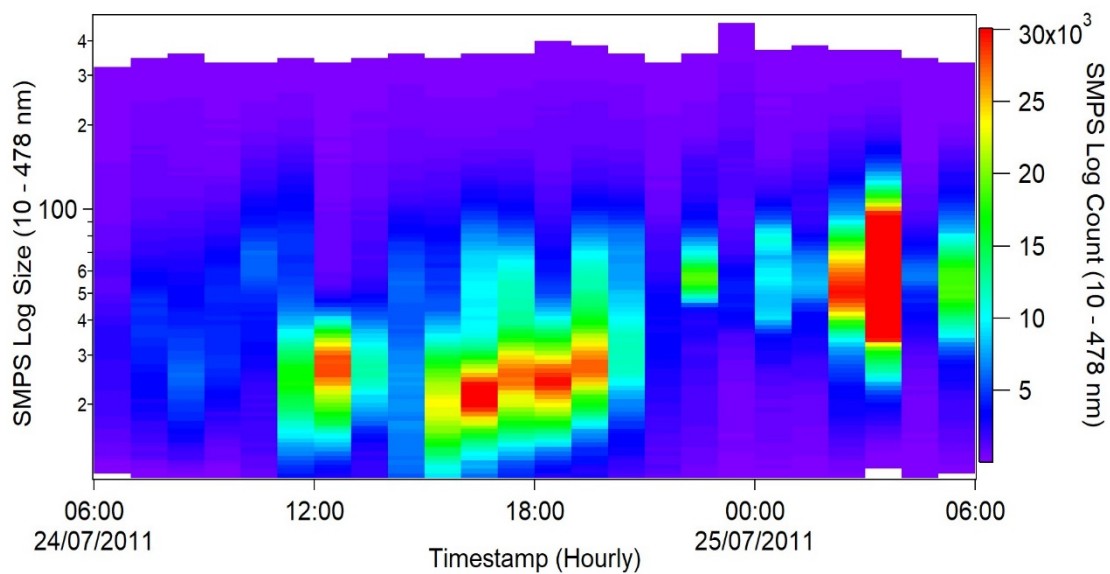

***Figure 6.* SMPS count data in the 10 - 478 nm range over 24-25 July, 2011**

Of most note in Figure 6 is the fact that during the high fluorescent particle count registered by the WIBS (~03.00 to 12.00 period) few particles were observed in the nanometre size regime appropriate to the SMPS detection system. In fact, the nano-particles only became measurable at ~12.00 and then carried on forming until ~18.00. The most intense events were not observable until ~03.00 on 25 July, some 15 hours after the fluorescent particle event began to decline.

A similar WIBS event was noted between 26[th] and 27[th] July. The drop-off in fluorescence counts coincided with an almost total extinction in visibility at the Roches' point monitoring station in the late afternoon and evening of 27[th] July. The relationship is shown in Figure 7.

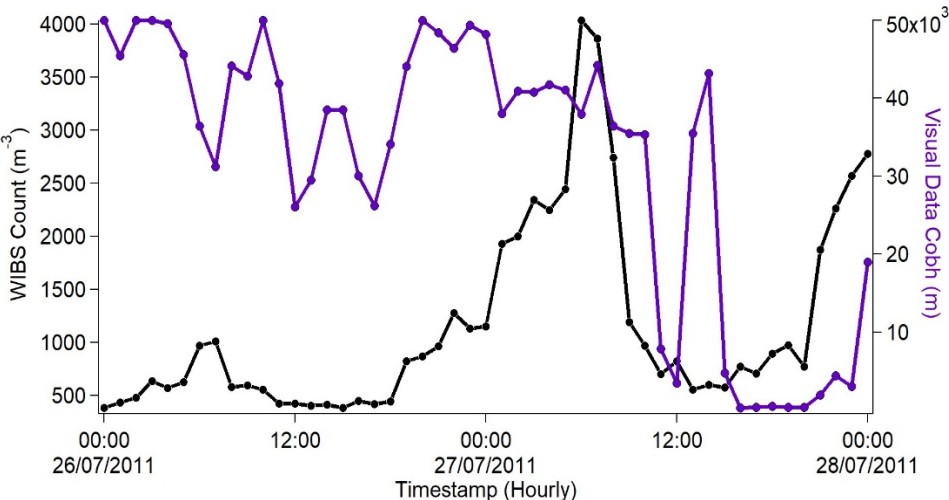

*Figure 7:* **Total fluorescence counts compared to visual data measurements (in m) as a function of time between 26<sup>th</sup> and 27<sup>th</sup> July 2011.**

Figure 7 also shows that few WIBS counts were measured on 26<sup>th</sup> July until ~23.00. A large continual increase then became apparent until 07.00-08.00 on 27<sup>th</sup> July followed by a rapid drop off until 12.00. A short-term increase in visibility was observed over the next hour and then rapidly declined again. The origin of this phenomenon is unknown but potentially could be due to a patch of shallow fog. By 14.00 the sea-fog formation event had begun. This type of WIBS count increase and decrease behaviour, between night and day when a sea-fog event occurred, was monitored on several occasions during the full campaign at Haulbowline Island. WIBS counts that increase and decrease in all fluorescent channels have been observed before, for example in a study of nightly inland fungal spore releases (O'Connor et al., 2015). However, the phenomenon observed here occurred at a coastline where the fluorescence signals were almost exclusively in the FL1 channel with no contribution from FL3. The signals also correlated with low-tide, sea kelp exposure and low wind speeds.

This behaviour is consistent with a build-up of fluorescent particles that are lost after sunrise and then, at some stage, after the fluorescent particles have transformed to non-fluorescent CCN, a sea-mist is formed. Particles much smaller than those measured as fluorescent can act as CCN in the development of a mist and so the size regime < 200 nm was also investigated.

SMPS measurements made over this time span revealed clear particle growth in the 10-100 nm size range beginning coincidentally (14.00) with the visibility decrease. A visualization of this data and an associated wind-rose plot are shown in Figure 8.

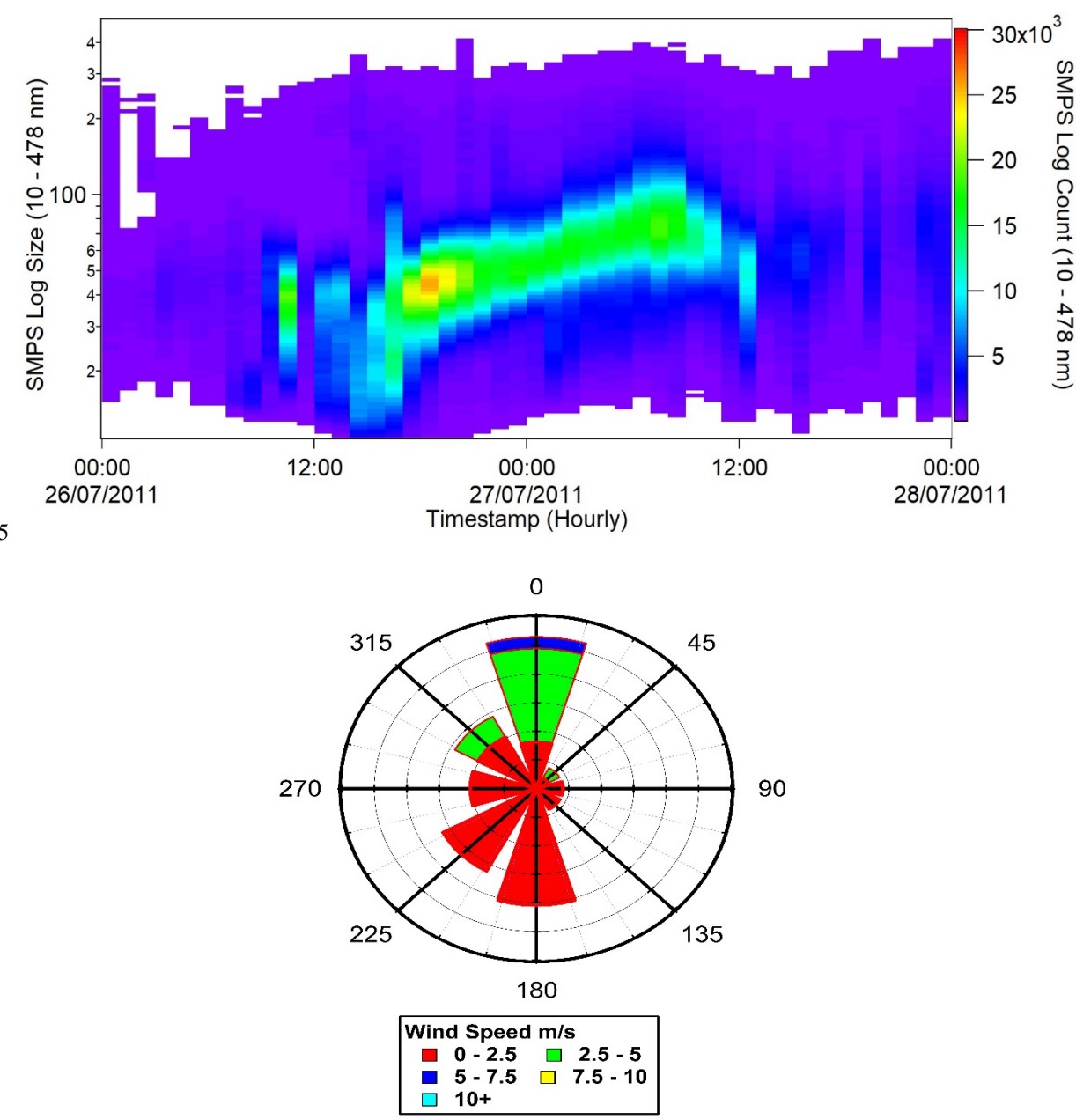

*Figure 8:* **Particle growth data obtained between 26th and 27th July with (A) an SMPS image plot; (B) a wind rose diagram illustrating the monitored wind speeds and directions.**

Figure 8(A) shows a plot of the growth of particles in the 10-100 nm size range between 26th and 27th July. The monitoring period began at ~14:00 on the 26th July with counts

<11,000 for the 20 – 50 nm size range. Within an hour the particle counts exceeded 11,000 and was followed by the intense growth (>20,000 counts) of particles with size ~40 nm by 17:00 on that day. Over the next 14-16 hours growth occurred leading to particles of size ~100 nm with counts 15,000-20,000. This period ended by 09:00 on the 27[th] July. The overall behaviour is fully indicative of particle nucleation and the 'banana' curve recorded is typical when such a process occurs. (Cheung et al., 2011). The wind rose plot shown in Figure 8 (B) indicates that the wind activity in the 15-hour time period was very low with speeds < 5 m/s providing conditions in which little particle dispersion would occur.

Both of the July periods discussed above experienced low tides of between 0.7 and 1.1 m (See supplementary data). The 16[th] July period coincided with a Full Moon and gave rise to a lowest monthly tide of < 0.5 m. This period was therefore analysed to bring together tide data, WIBS and SMPS counts, solar activity, visibility and meteorological data to illustrate the time-line of the phenomena observed during the July campaign at Haulbowline Island. The information is presented in Figure 9 (A), (B), (C), (D), (E).

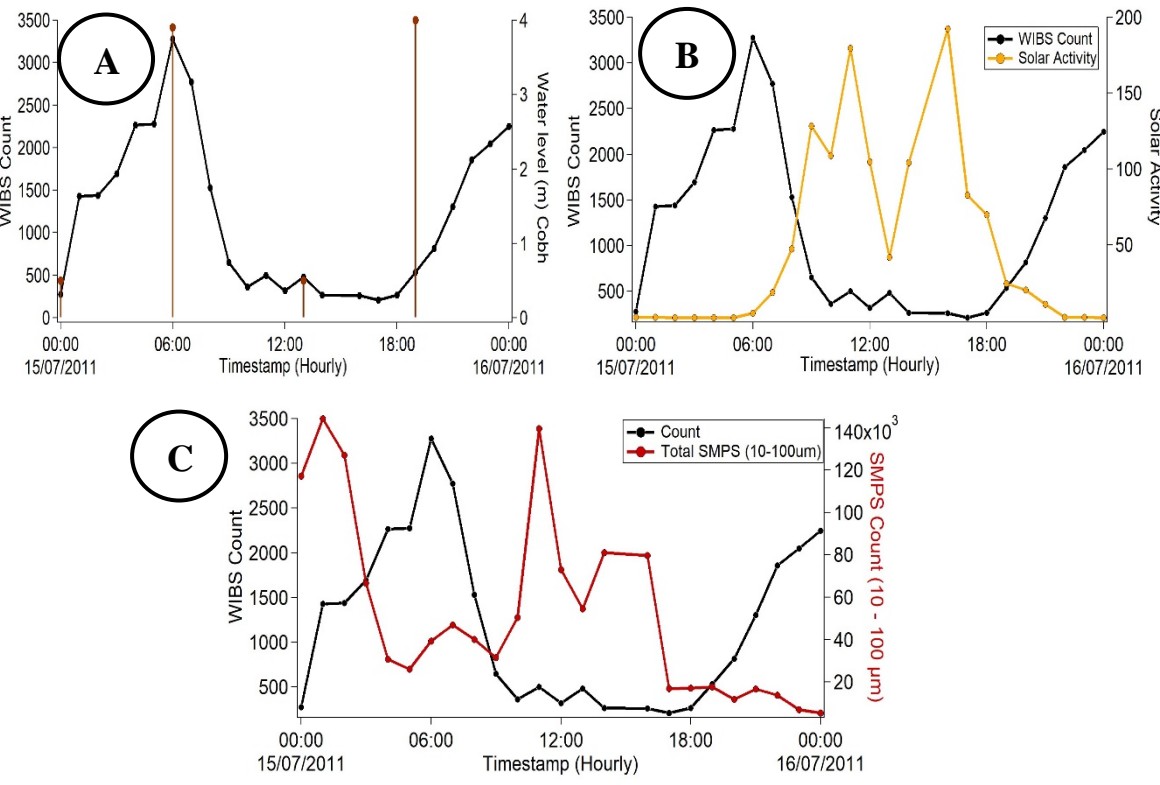

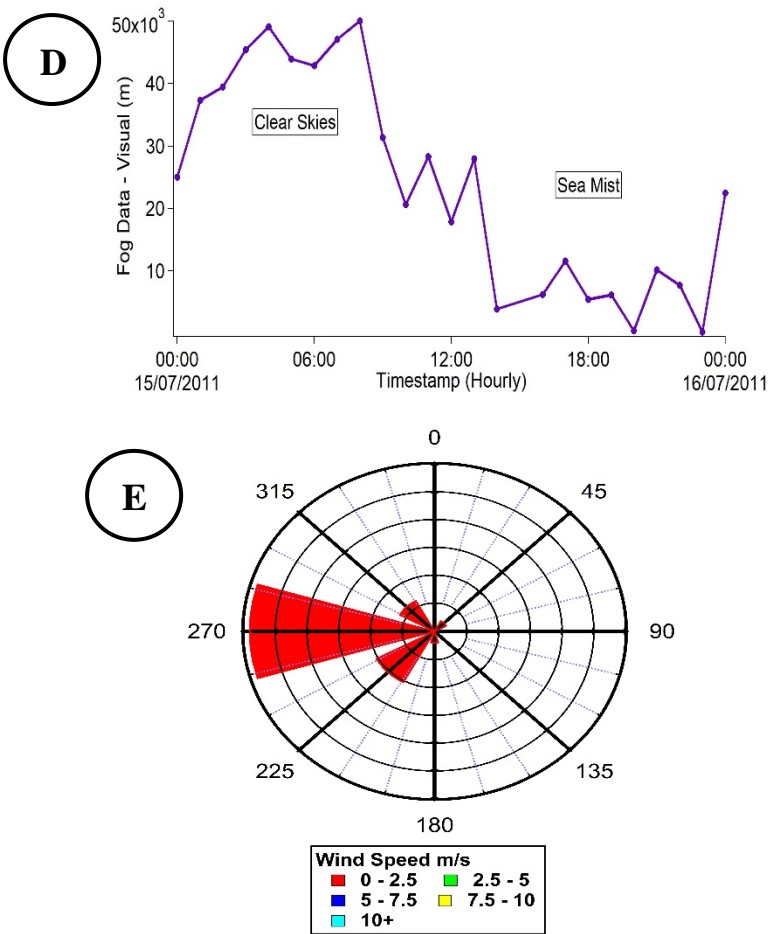

*Figure 9:* **One-day timeline trend between 15<sup>th</sup> July (00:00) and 16<sup>th</sup> July (00:00). (A) WIBS count vs Tidal data; (B) WIBS count vs solar activity in watts per square meter (W/m²); (C) SMPS data vs WIBS count; (D) Visual range data; (E) Windrose plot for wind direction and speed during 15/16 July.**

In summary, at approximately midnight a rapid increase in fluorescence signal was observed. This result coincides with a period when very low tides were present at the Haulbowline site and *Laminaria* kelps were exposed to the atmosphere. After sunrise the WIBS counts decreased and were low throughout the full hours of daylight. The counts only returned when solar activity was low. The time profile of the SMPS counts appear to anti-correlate with the WIBS fluorescent counts as shown in Figure 9 (C). The smaller particles reached their lowest levels when the WIBS counts increased and coincided with the beginning of the evening period when much less daylight was present. From the visibility measurements it is clear that a sea-mist began to form at about noon when the SMPS counts began to be registered. A fog developed late afternoon and was present until

about midnight. This unique set of timeline events gives a clear linkage between the formation of fluorescent particles of 2-6 μm size to a sea mist/sea fog after about 12 hours. These observations were made at a time when low wind speeds (0.2-2.3 ms$^{-1}$) prevailed.

The WIBS results show that the fluorescent particles detected were potentially not of biological nature due to the lack of FL3 fluorescence (Healy et al., 2012a; Hernandez et al., 2016; O'Connor et al., 2014; Toprak et al., 2013) but clearly, from previous studies of CCN at coastlines, it is possible that they might be related to the production of iodine, which fluoresces in the vapour-phase along with timely release with exposed sea kelp at

low tides. Hence a series of laboratory experiments using the WIBS was performed to assess any potential role of iodine in the development of the sea mists in July, 2011 at Haulbowline Island. It should be not that other non-biological particles have been seen to be fluorescent. Mineral dust was also considered as a potential source of fluorescent particles. In fact, studies have shown that fluorescent mineral dust can contribute up to

~10% of the total measured (Toprak and Schnaiter., 2013). However, in this study, using the FL1 and FL3 channel filters this dust artefact was removed. A lack of FL3 fluorescence signals in this study rules out the presence of mineral dust because it weakly fluoresces in the FL1 and FL3 detection ranges and therefore is considerably weaker than biofluorophore signals (Toprak and Schnaiter, 2013; Pohlker et al., 2012). Polycyclic Aromatic

Hydrocarbons (PAH's) could also be considered a potential interference to the measurements made here due to their highly fluorescent nature but since they are largely present on the surface of soot particles which generally exist in submicron sizes, detection by the WIBS is unlikely unless oil droplets were present. The complex chemical environments associated with soot particles can also lead to fluorescence quenching of

PAH's (Pohlker et al., 2012). Humic-like substances (HULIS) and secondary organic aerosols (SOA) have also been indicated as potential interference signals in the literature (Pohlker et al., 2012).

*3.2 Laboratory experiments*

To help interpret the results obtained in the field campaign, a series of fluorescence and UV absorption measurements were performed on iodine dispersions in water. The laboratory set-up to investigate these aerosols using a WIBS-4A as a monitor for fluorescence is shown in Figure 1.

As mentioned earlier, the conventional forced trigger procedure to set a baseline for the WIBS fluorescence signals was performed and gave the following baseline values, FL1: 134, FL2: 20, FL3: 25. Table 1 shows the percentage values measured for each fluorescence channel filter (A, B, C etc) in relation to the total fluorescence particle count for the results obtained in the chamber experiments.

*Table 1:* **Percentage value of each fluorescence channel filter in relation to the total count.**

| Name | %A | %B | %C | %AB | %AC | %BC | %ABC |
|---|---|---|---|---|---|---|---|
| Saltwater or saltwater with sublimed $I_2$ | 4.02 | 2.42 | 0.03 | 0.01 | 0.94 | 0.02 | 0 |
| MilliQ water droplets | 3.46 | 3.51 | 0.03 | 0.04 | 0.7 | 0.01 | 0 |
| Iodine water solution | 0.84 | 6.69 | 5.85 | 0.04 | 0.04 | 1.56 | 0.63 |
| MilliQ water droplets with sublimed $I_2$ | 48.61 | 0.88 | 0.38 | 13.71 | 0.93 | 0.41 | 1.18 |

The most striking observation from the data presented in Table 1 is that the only channel filter giving substantial (~50% contribution) to the total count is A (*i.e.* excitation at 280 nm, emission 310-400 nm, the FL1 channel) for the experiments in which MilliQ water droplets were co-introduced with sublimed iodine. The next largest contribution was observed from the same water/sublimed iodine dispersion in the AB dual filter channel (*i.e.* excitation at 280 nm, emission 310-400 nm, the FL1 channel; excitation at 280 nm, emission 420-650 nm, the FL2 Channel). All other contributions were small (<5%) for all other chamber introductions including iodine vapour and aerosolized iodine/water solutions. Iodine vapour fluorescence was not measurable because it does not provide particles which are necessary for detection by the internal diode laser of the WIBS. If the laser does not detect a particle of the specified size, the flash lamps are not triggered and therefore no fluorescence is detected because no excitation occurs. One possibility is that a non-fluorescent particle could be detected, which activates the flash lamps and a fluorescent signal from the surrounding iodine vapour is detected. However, with the chamber being pumped down for 30 minutes before each experiment, no particles of that size can interfere.

The next important observation is that little to no FL C channel fluorescence (i.e. excitation at 370 nm, emission 420-650 nm, the FL3 Channel) was detected in any experiment. These results taken together indicate that the fluorescing species absorbs at 280 nm and, at best, weakly emits in the 420-650 nm region. This behaviour was very different from that observed when dispersed pollen and spores were introduced into a chamber. (Healy et al., 2012a; O'Connor et al., 2014; Toprak et al., 2013).

It is also noteworthy that sublimed iodine/saltwater mixture results were identical to those using saltwater alone giving very low fluorescence contributions in all channels. Indeed, in both cases, the FL A filtered channel represented about one tenth of the population measured to be fluorescent for their sublimed iodine/MilliQ water counterparts. A possible explanation for this observation would be the quenching effect of sea salt chloride ions or iodide ions on the iodine vapour (Chmyrov A et al., 2010; Martin et al., 1997) or even the formation of iodine chloride ions as discussed below.

Droplet size *vs* count measurements were also made for the various dispersions to compare with the results obtained in the Haulbowline field campaign. Figure 10 shows these results: (A) from the field study; (B) the saltwater dispersions; (C) the sublimed iodine/MilliQ water aerosols. It is clear that the size distributions for the non-fluorescent water/salt water droplets (0.5-1.5 μm) are in good agreement with the data obtained in the field. Most importantly, the sizes of the dispersions fluorescing in the FL1/FLA range are close to identical for field and laboratory displaying a maximum 2-2.5 μm in each case. However, the Haulbowline measurements indicate that reduced fluorescent counts are measured up to droplet sizes of ~6 μm whereas the laboratory distributions only reach about 3-3.5 μm. Generally, aerosolised sea spray in coastal regions have been recorded in the size range measured in these experiments *i.e.* from 0.5 to 5 μm for "film" droplets, which are formed from surface bubble bursting (Andreas., 1998).

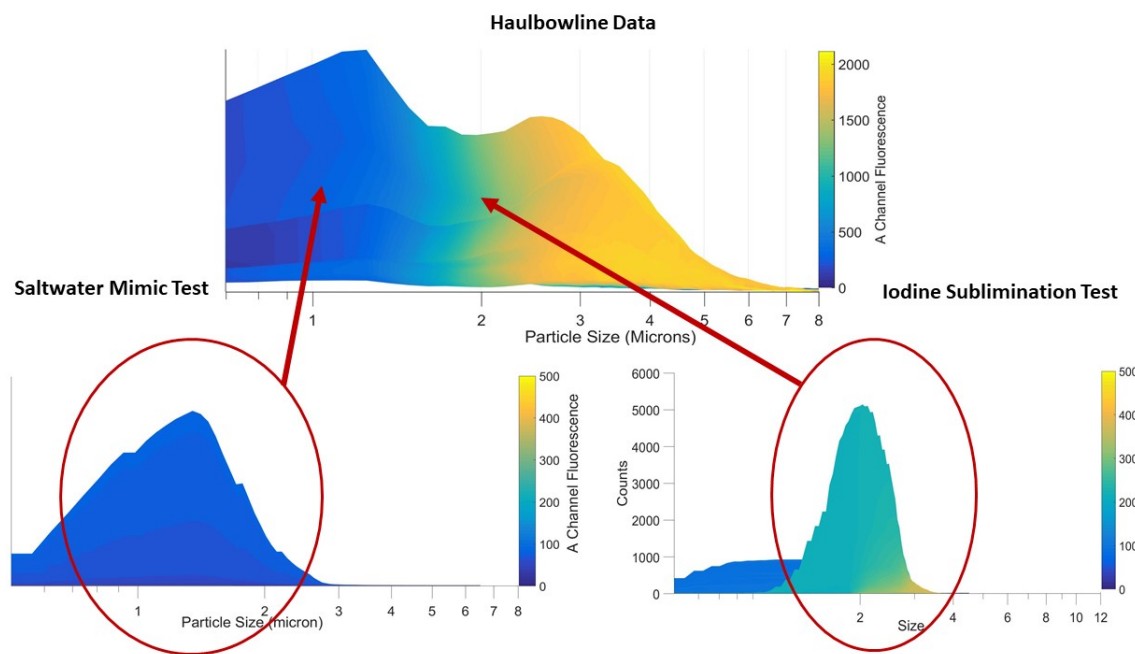

*Figure 10.* **Comparison between droplet/particle size vs FL1/FLA data obtained in (A) the Haulbowline campaign; (B) saltwater aerosol chamber experiment; (C) sublimed iodine/MilliQ water aerosol chamber experiment.**

Although the field/laboratory comparison experiments give a good explanation for the observations made during the Haulbowline campaign, various questions do arise from the WIBS results obtained in the laboratory. For example, why do aerosolized iodine solutions appear to not fluoresce and why does the FLA/FL1 channel dominate the only dispersion that fluoresces strongly, namely, sublimed iodine/MilliQ water?

It has long been known that iodine vapour displays a pronounced orange-yellow fluorescence when excited by visible light whereas its solutions and solid form do not. (Lommel, E., 1883). In fact, solutions of iodine in water can rapidly form the trihalide ion $I_3^-$ and in saltwater $I_2Cl^-$ can be produced as well as $I_3^-$ ions over time. UV-Vis spectra of a number of solutions were measured in this study and were in good agreement with previously published work. (Alizadeh et al., 2012, Leblanc et al., 2006; Meyerstein et al., 1962; O'Driscoll et al., 2008). Additional iodine-chlorine trihalides can also be formed (Kazantseva et al., 2002; Meyerstein et al., 1962; O'Driscoll et al., 2006). The absorption wavelength maxima corresponding to the various species are given in Table 2. The $I_3^-$ peak at 352 nm was not present at all in cold saltwater mimic samples and only in trace amounts

(<5% of iodine and water mix) in heated samples while the peak at 288 nm largely remained. Work reported by Alizadeh et al (2012) showed an $I_2$ peak at 450 nm but also a small absorbance at 250-290 nm. This result shows the possibility of $I_2$ absorbance at ~280 nm. The absorption spectra recorded for each sample is available in the supplementary material.

*Table 2.* **Absorption wavelength maxima for iodine solutions in water and sea salt solutions**

| Sample (Absorption wavelength) | $I_2$ + $I_3^-$ (288nm) | $I_2$ (450nm) | $I_3^-$ (352nm) | $I_2Cl^-$ (248nm) | $I_2Cl^-$ (437nm) |
|---|---|---|---|---|---|
| Iodine & water | Yes | Yes | Yes | No | No |
| Iodine & saltwater | $I_2$ -Yes $I_3^-$ (<5%) | Yes | Weakly (<5%) | Yes | Yes |

These absorption spectra results mean that the two WIBS filtered xenon flash lamp UV sources (280 nm and 370 nm) should excite iodine/water solutions whereas the saltwater counterpart would only be efficiently excited by the 280 nm source. However little fluorescence was observed in the chamber experiments for either of the aerosolised *solutions*. No reports of trihalide ion fluorescence in solution have been reported previously and any molecular iodine cannot be detected by WIBS.

It is known for $I_2$ in vapour form that the transition probability of the D-X system at 280 nm is <1 x $10^{18}$ cm$^2$. This value is about eighteen times smaller than the peak absorbance at 203 nm (Saiz-Lopez et al., 2004). However work by (Liu et al., 2004) show that if $I_2$ is complexed with different organic solvents such as toluene, a strong absorption band is present at ~280 nm due to the molecular $I_2$ UV spectrum red-shifting. A similar interaction can be envisioned for iodine bound to the surface of water droplets rather than being present in the vapour phase or simply dissolved inside a droplet.

The results in Table 1 indicate that molecular I₂, dissolved in water does not give rise to fluorescence signals. Hence the likely explanation for the observed FL1 fluorescence in the water droplet/iodine vapour experiments is that I₂ molecules can be adsorbed to water droplet surfaces. This physical condition would not necessarily apply to the aerosols in

which iodine is initially dissolved in water. In other words, adsorption of the vapour onto the droplet surface could give rise to much more "free", unsolvated forms of the molecular iodine.

*3.3 Mechanisms to explain the field and laboratory results*

As outlined above, the WIBS fluorescence and non-fluorescent particle data obtained during the Haulbowline field campaign as shown in Figure 2 and Figure 10 did not resemble any prior sizing datasets obtained in airborne PBAP detection. Furthermore, the fact that significant fluorescence signals could only be recorded in the FL1 channel differentiated the campaign from any that had been previously directed toward real-time

monitoring of PBAP/bioaerosols using Light Induced Fluorescence (LIF). Other campaigns involving the use of the WIBS included locations ranging from composting sites (O'Connor et al., 2015) to tropical forests (Stanley et al., 2011; Valsan et al., 2016) and urban areas (Huffman et al., 2010) have been performed but none have been carried out at a coastline.

Any fluorescence signals from chemical aerosols like Secondary Organic Aerosol (SOA) measured to date by WIBS in field campaigns are generally associated with small particles and have never been registered with sizes >5 μm. Such chemicals also fluoresce across all three channels. Hence, an alternative source of the fluorescence signals at a coastline is

iodine because its vapour fluoresces and is known to be released by kelps during low tides. Organic compounds adsorbed to sea-salt might represent a further possibility but would be less likely because of fluorescence quenching, as indicated by the chamber studies described above using iodine vapour and sea-salt.

In fact, the laboratory chamber results performed in this study centred on iodine vapour and various iodine aerosol dispersions showed that only the sublimed iodine/MilliQ water dispersions reproduced almost exactly the bimodal size/fluorescence distributions monitored by WIBS on several occasions in July 2011 at Haulbowline Island. This unique behaviour can be explained by considering a role for pure water droplets to act as

interfacial, surface "carriers" of molecular iodine rather than including it as fully solvated molecular iodine or as trihalide ions. Salt water counterparts did not fluoresce to any great extent perhaps because excess chloride ions on a droplet surface can quench iodine fluorescence by rapidly forming surface tri-halide ions or because they reduce attractive forces that stabilise the molecular iodine carried by the surface.

In view of the WIBS laboratory and field results outlined above a small but potentially important adaption of the currently accepted mechanism linking iodide ion production from sea kelp to the presence of small airborne $I_xO_y$ particles is proposed (Küpper et al., 2008; O'Dowd et al., 2002). This hypothesis is centred on the night-time formation of relatively stable surface "chaperone" $I_2 \cdot (H_2O)_x$ complexes being emitted from coastal sea spray releases.

Hence in conditions with no light and at low tide, iodide ions released from coastal sea kelp interact with tropospheric ozone to form molecular iodine (Huang et al., 2010) In daytime the first order rate of destruction of $I_2$ due to photolysis is 0.14 s$^{-1}$ (*i.e.* an atmospheric lifetime of only 7 s) (McFiggans et al., 2004). The other main routes for its destruction proposed are by rain-out and in aerosols, which would lead to facile dissolution (Baker et al., 2001). However, it is proposed from the results of this study that the emitted $I_2$ can interact with sea spray to form $I_2 \cdot (H_2O)_x$ complexes. Their involvement would help to increase the atmospheric lifetime of molecular iodine substantially and also promote its night-time transportation to some distance away from the coastal source. Work by Galvez et al (2013) suggests a theoretical weakly bound complex of one $I_2$ molecule with one $H_2O$ molecule (14 kJ mol$^{-1}$). However, the current study likely addresses cases when several $I_2$ molecules bind to one or more water droplets in a surface adsorption mechanism.

When the sun rises, photolysis of the interfacial $I_2$/water droplets would lead, in the normal way, to the formation of iodine radical species, which can then interact with ozone to give IO radicals. These may then agglomerate to yield $I_xO_y$ complexes like the stable oxide $I_2O_5$ that can subsequently act as CCN (Cloud Condensation Nuclei).

The proposal that $I_2 \cdot (H_2O)_x$ intermediates are involved in the formation of $I_2O_5$ clusters is further supported by the recent discovery that both iodine oxoacids and iodine oxide vapours are effective and efficient precursors to airborne coastal particles. (Sipila et al.,

2016). Thus, in the laboratory iodic acid ($HIO_3$) may be produced from the reaction between molecular iodine and water in the presence of a strong oxidiser such as hydrogen peroxide, chlorine or even ozone. However, the most important atmospheric day time oxidant is the OH radical and so its potential reaction with the $I_2 \cdot (H_2O)_x$ complexes to give airborne $HIO_3$ also deserves future consideration.

In fact, a recent report, which outlines evidence for some coastal aerosol particle formation being due, in part, to the sequential addition of $HIO_3$ indicates that at Mace Head, Ireland, the production of the oxo-acid has been shown to begin at sunrise reaching a maximum at noon (Sipila et al., 2016). It should be noted that the measurements were made during the August to October period. During that time, the iodic acid is reported to appear at ~13.00 with the iodine oxide clusters peaking at ~15.00 pm due to later sunrise. It is highly likely IO radicals were present just before these clusters formed.

Since the sun rises at an earlier time in July, it should be expected that the reported $I_xO_y$ particles measured in this study by the SMPS peak earlier as found at 13.00. Therefore, the IO radicals would be present in the gas-phase even earlier before building the particles measured at 13:00. This phenomenon, or the intermediacy of $HIO_3$, could also account for the ~3 hours lag time observed between the WIBS peak decrease for the $I_2 \cdot (H_2O)_x$ species to the SMPS count increase.

The nucleating properties of $I_2O_5$ particles should be expected to lead to the formation of sea-fogs/-mists particularly when little wind is present as observed in the field study presented here. Finally, it is worth noting that iodine oxides have been detected just after sunrise in previous studies and that no traces of IO were detected at night time leading to the conclusion that photosensitive reserves of some sort must be present (Zingler et al., 2005). $I_2 \cdot (H_2O)_x$ complexes could certainly be considered as potential candidates for such reservoirs.

## 4 Conclusion

For many years, sea-salt particles were accepted as the only types of CCN that drive sea-fog formation but over the last ten years or so small iodine oxide particles released from atmospherically-stressed kelp have also been identified as a source. However, there have

been no previous field measurements that have provided a direct time-line link between molecular iodine release, particle formation and sea-fog formation.

The dual field and laboratory study presented here provides a possible real-time profile for
the formation of coastal sea mists and fogs as observed on several occasions in July 2011 at Haulbowline Island, Co. Cork. The fluorescence results combining intensity measurements with sizing information provide a potential new mechanism for coastal sea-fog formation involving the atmospheric formation and dispersion of molecular iodine as $I_2 \cdot (H_2O)_x$ surface complexes on sea-spray droplets. Of mechanistic interest is the fact that
molecular iodine included into (rather than on) water droplets do not appear to fluoresce by measurements using WIBS instrumentation. However, the most important finding is that the study indicates a previously unsuspected stabilizing transport mechanism for iodine in the marine environment. Hence the stabilization of the molecular form would allow its more extensive distribution throughout the troposphere before photolysis.

**Author contributions:**

Shane M. Daly, David A. Healy, Jovanna Arndt, Eoin J. McGillicuddy, Patrick Feeney and Michael Quirke carried out Field measurements and operation of the monitoring station. David J. O'Connor and Shane M. Daly carried out analysis of results and contributed to
preparation of manuscript. Stig Hellebust designed MATLAB data analysis tool, advised on data analysis and contributed to manuscript editing. John Wenger contributed to manuscript editing. John Sodeau was Principal Investigator and prepared the manuscript with contributions from all co-authors.

**Acknowledgements:**

Financial support from the Irish Environmental Protection Agency through the Climate Change Research Programme 2007-2013 for the project "Analyses of the Development and Occurrence of Biological and Chemical Aerosols (BioCheA)" (project no. 2007 CCRP Project 4.4.6.b) is gratefully acknowledged.

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
