# Peer review of "Investigation of coastal sea-fog formation using the WIBS (Wideband Integrated Bioaerosol Sensor) technique"

_Atmospheric Chemistry and Physics, 2018_

## Referee Comment (RC1) · Anonymous Referee #1 · 27 Sep 2018

**General comments**

Daly et al present a very interesting study of particle formation in a coastal environment. Using a wideband integrated bioaerosol sensor (WIBS), a scanning mobility particle sizer (SMPS) and laser scattering visibility observations, they found evidence linking molecular iodine release and particle formation with sea-fog formation. The paper addresses a relevant scientific question within the scope of ACP, presents a novel concept and reaches substantial conclusions. I recommend publication after some minor changes.

**Specific comments**

[Figure]

While the timeline link between molecular iodine release at low tide, iodine oxide formation (in daytime) and particle formation is well stablished (references cited in the paper itself), the study links these two with the formation of coastal mist for the first time. I would then recommend a slightly more nuanced statement about the novelty of this 3-step time-line in the abstract and conclusions. What is certainly new and exciting about the study is the field observation of I2 adsorbed onto water droplets. This appears to be a previously unknown stabilizing transport mechanism for the dispersal of I2 in the marine environment. This finding is supported by targeted laboratory experiments demonstrating that the non-biological WIBS signals observed in the field are in fact consistent with the adsorption of I2 on the surface of nebulized pure water droplets.

The absence of fluorescence for nebulized seawater in the presence of iodine vapour may well be explained by the formation of trihalide ions as the authors suggest. It would be interesting to know, though, whether the saline solutions prepared for the lab experiments have similar iodide and chloride concentrations to those expected in sea spray, and whether more diluted concentrations would have resulted in increased signal in the FL1 channel. Also, whether changing the residence time of the nebulized sea salt solution in the chamber could have shown some evidence of the kinetics of the removal of the adsorbed I2 molecules at the surface. I any case, the characterisation of the sea salt solution should be included in section 2.3. I would like to see also some information about the pressure, temperatre, flow rate, and residence time conditions in the aerosol dispersion chamber.

The absence of fluorescence when only iodine vapor is admitted in the laboratory chamber is very interesting, but I don't find very convincing the argument given by the authors to explain this point. The WIBS-4 instrument is optimized for particle detection, but in its configuration (http://www.dropletmeasurement.com/wideband-integrated-bioaerosol-sensor-wibs-neo) I don't find any obstacle for the detection of gas-phase fluorescence (perhaps the authors could comment more on that to make it

clearer). I2 molecular iodine presents a strong absorption feature in the 170–210 nm spectral range (the Cordes bands, D-X system), with peak absorption cross section of 2E−17 cm2 molecule−1 at 188 nm (Myer and Samson, 1970; Roxlo and Mandl, 1980). After absorption of VUV and middle UV photons, fluorescence from the D ion-pair state back to the ground state exhibits an ordinary bound to bound spectrum together with a bound to free diffuse quantum interference spectrum (the McLennan bands) (Tellinghuisen, 1974; Exton and Balla, 2004). Concurrently, a significant fraction of the initial D state population is collisionally transferred to the D' state at increasing buffer gas pressure, resulting in fluorescence in the D'-A' band at 340 nm. Since the transition probability of the D-X system at 280 nm is small (absorption cross section of 6E−19 cm2 molecule−1 (Saiz-Lopez et al., 2004)), this may explain why I2 is not observed in the gas phase. On the other hand, complexation with water may red-shift the absorption spectrum. E.g. the peak of the I2 VUV band shifts to 203 nm in aqueous solution (Kireev and Shnyrev 2015). This would be a plausible explanation as to why 280nm-pumped fluorescence in the 310 - 400 nm range can be observed when I2 is complexed with water and not in the absence of water droplets. In feel that some spectroscopic discussion in this sense is pertinent.

Section 2.3 mentions that fluorescence spectra of the solutions were investigated using a Shimadzu RF-6000, but the results of these investigations are not reported. I also find that the solution absorption data is presented in a rather schematic way and that it would be more informative to show absorbance spectra (perhaps in the supplementary material) to demonstrate how efficient is the absorption of the two WIBS wavelengths.

The suggested link between I2.(H2O)x and HIO3 is a problematic one. Sipila et al. 2016 did observe HIO3 and molecular cluster formation in their laboratory experiments, but in the presence of water vapor (no nebulized water droplets). Daly et al. do not report WIBS measurements of a mixture of water and iodine vapor, but it is known that I2 and H2O form a weakly bound complex (Galvez et al. 2013), and under atmospheric conditions only a residual amount of I2 would be complexed with H2O. The presence

of I2.(H2O)x in the laboratory experiments of Sipila et al. is therefore unlikely. Furthermore, HIO3 increase was observed by Sipila et al. in the field at daytime, well after noon, while Daly et al show that I2(H2O)x is a night time reservoir which disappear quickly after sunrise.

**Minor comments**

Page 6, line 20: 0.5-20 m. do you mean micron?

Page 11, lines 2 and 3: it looks like all these size ranges should be microns rather than meters.

All figures. In general, the legends and axis labels are too small and difficult to read, especially in multi-panel figures.

Figure 9: some of the tidal height data is missing: the 6:00AM and 18:00 tidal values are not shown (as opposed to figure 1 in the supplementary information).

––––––––––––––––––––––––––––––––

---

## Referee Comment (RC2) · Anonymous Referee #2 · 18 Oct 2018

Daly et al. submitted a manuscript for review titled "Investigation of coastal sea-fog formation using the WIBS (Wideband Integrated Bioaerosol Sensor) technique." The manuscript observes and analyzes data from a WIBS-4 that was deployed in Haulbowline Island, Cork Harbor during July and September 2011. The author states that size and fluorescence profiles indicated that the origin of the signals were not biological in nature. A second single-particle fluorescence spectrometer, a WIBS-4A system, was used for complementary laboratory studies to help explain field results. The laboratories studies are thought to explain a possible mechanism seen by the WIBS-4 system deployed in the Cork Harbor, which suggests the idea of the adsorption of molecular iodine onto water droplets to form $I_2(H_2O)_x$ complexes. The study elutes to an un-

suspected stabilizing transport mechanism for iodine in the marine environment and provides the first real-time link between molecular iodine release, particle formation and sea-fog formation. In general, I support the publication of this manuscript in some form, however I believe there are several comments that need to be addressed before consideration. There are several peer-reviewed publications that have explored different analysis strategies for the WIBS that were not mentioned, nonetheless should be considered and discussed. I list some suggestions for specific additions below, including some possibilities for added discussion and some suggestions.

General comments:

Section 2 Methodology: In general, I think Section 2 needs a more detail on A) the laboratory experiments that took place using the WIBS-4A instrument and B) the preparation for the data for both the field analysis and complementary laboratory studies. Since this manuscript is the first evidence of using real-time fluorescence spectrometers to observe the link between molecular iodine release, particle and sea-fog formation, I think it is crucial for the methodology to be written so that it can be repeated and further explored, specifically in Section 2.3. There has been several studies on the preparation of WIBS data, before subsequent analysis and how this may change the overall observations, e.g. Gabey et al., 2010, Perring et al., 2015, Savage et al., 2017 and Savage et al., 2018, and references there-in. All the mentioned studies look into fluorescence thresholding and how this may result in the efficiency at which the WIBS can discriminate between biological and non-biological particles. From my understanding the author uses the FT signals as the fluorescence threshold, and compares results to what was seen in the Hernandez et al., 2016 publication, however this publication uses the default FT + 3sigma threshold. I suggest the author goes into more detail on why they chose to use the threshold they did, and what implications this may have on their results. Some of this extensive detail may belong in the discussion, however how the author 'prepares' the data for both the laboratory and field study should be more explicit (e.g. size calibration information, fluorescence calibration information,

the fluorescence threshold chosen- whether it is the average, median, etc.).

Section 2.2 Field Instrumentation: On page 6 lines 21-24, the author states that both the WIBS-4 and the WIBS-4A units were identical in terms of functionally- this is strong statement. Studies including Robinson et al., 2017, Savage et al., 2017, and Tobias et al., 2018 explain the current hurdles to when comparing data from two different WIBS units (or fluorescence spectrometers). Hernandez et al., 2016 uses two different WIBS units in his studies, and it can be seen first-hand the differences in fluorescence signals produced by two different instruments observing and measuring the same particle type. The Robinson et al., 2017 publication provides a procedure for calibrating the different WIBS channels for the inter-comparison of WIBS data. Can the author please comment on whether such calibration was done? Where the PMT voltages measured for each WIBS unit?

Page 11, lines 5- 20: In general, I think this section needs more discussion in regards to the suggested publications and their analysis strategies - Gabey et al., 2010, Perring et al., 2015, Savage et al., 2017 and Savage et al., 2018. It is not clear what the author means by stating " Unusually, fluorescence signals were mainly measurable in the FL1 channel. FL2 registered little emission above threshold as illustrated in Figure 4, which shows plots of size/AF data as a function of the FL1 and FL2 channels. (FL3 showed no fluorescence). The larger size feature (2-6 um) consisting of highly fluorescent solid particles/droplets but only in the FL1 channel represents a behaviour that has not been observed previously in any WIBS field campaign. Hence fungal spores, certain pollen and bacteria as large as 2 um (Hernandez et al., 2016) can be found in the 2-6 um size regime but are fluorescent in all channels because of their amino acid, tryptophan and NAD(P)H contents". The excellent study by Hernandez et al., 2016 provided the first extensive characterization of the WIBS using various biological particles. It is important to keep in mind the many particles, e.g. pollen and fungi, are larger than 6 um in diameter and one's aerosolization technique may influence the size distribution observed. Savage et al., 2016 showed that aerosolization of pollen using turbulence

created with a stir bar resulted in fragmented pollen. On another note, the author compares fluorescence signals in this study with those observed in Hernandez et al., 2016, however this cited publication uses a different fluorescence threshold. Studies have shown that biological particles may not have FL3 fluorescence characteristics when observed by the WIBS, some of these particles include various bacteria and fungi.

Several studies suggest there are non-biological, fluorescent particles that may be interferences when discriminating between bio vs. non-biological particles, and even different particle types (Huffman et al., 2010, Pohlker et al., 2015, and Savage et al., 2017, and references there-in). Can the author comment on these possible interferences, and if these substances were taken into consideration during their field analysis?

Minor Comments: Figure 4: The y-axis legend is unclear- is it the number of particles the succeed the threshold in all three channels? Figure 3: I suggest logging the x-axis. The y access legend is unclear- see comment for Figure 4. Page 11, lines 1-4 "It is clear, from the data shown in Figure 3, that a bimodal size distribution was recorded with: (i) a highly fluorescent, broad feature observed between ~2-6 $\mu$m, peaking at ~2.5 $\mu$m; (ii) a much narrower peak in the size regime <1.5 $\mu$m that represents non-fluorescent particles." Are these particles truly non-fluorescent or just 'weakly' fluorescent? It seems based on the gradient of the color legend in Figure 3 that most of these particles exhibit fluorescence that are indeed over the thresholds stated in page 10 line 7.

References: Huffman, J. A., Treutlein, B., and Pöschl, U.: Fluorescent biological aerosol particle concentrations and size distributions measured with an Ultraviolet Aerodynamic Particle Sizer (UVAPS) in Central Europe, Atmos. Chem. Phys., 10, 3215–3233, https://doi.org/10.5194/acp-10-3215-2010, 2010.

Gabey, A. M., Gallagher, M. W., Whitehead, J., Dorsey, J. R., Kaye, P. H., and Stanley,

W. R.: Measurements and comparison of primary biological aerosol above and below a tropical forest canopy using a dual channel fluorescence spectrometer, Atmos. Chem. Phys., 10, 4453–4466, https://doi.org/10.5194/acp10-4453-2010, 2010.

Pöhlker, C., Huffman, J. A., and Pöschl, U.: Autofluorescence of atmospheric bioaerosols – fluorescent biomolecules and potential interferences, Atmos. Meas. Tech., 5, 37–71, https://doi.org/10.5194/amt-5-37-2012, 2012.

Perring, A. E., Schwarz, J. P., Baumgardner, D., Hernandez, M. T., Spracklen, D. V., Heald, C. L., Gao, R. S., Kok, G., McMeeking, G. R., McQuaid, J. B., and Fahey, D. W.: Airborne observations of regional variation in fluorescent aerosol across the United States, J. Geophys. Res.-Atmos., 120, 1153–1170, https://doi.org/10.1002/2014JD022495, 2015

Robinson, E. S., Gao, R.-S., Schwarz, J. P., Fahey, D. W., and Perring, A. E.: Fluorescence calibration method for single-particle aerosol fluorescence instruments, Atmos. Meas. Tech., 10, 1755– 1768, https://doi.org/10.5194/amt-10-1755-2017, 2017.

Savage, N. J., Krentz, C. E., Könemann, T., Han, T. T., Mainelis, G., Pöhlker, C., and Huffman, J. A.: Systematic characterization and fluorescence threshold strategies for the wideband integrated bioaerosol sensor (WIBS) using size-resolved biological and interfering particles, Atmos. Meas. Tech., 10, 4279-4302, https://doi.org/10.5194/amt-10-4279-2017, 2017.

Savage, N. J. and Huffman, J. A.: Evaluation of a hierarchical agglomerative clustering method applied to WIBS laboratory data for improved discrimination of biological particles by comparing data preparation techniques, Atmos. Meas. Tech., 11, 4929-4942, https://doi.org/10.5194/amt-11-4929-2018, 2018.

Könemann, T., Savage, N. J., Huffman, J. A., and Pöhlker, C.: Characterization of steady-state fluorescence properties of polystyrene latex spheres using off- and online spectroscopic methods, Atmos. Meas. Tech., 11, 3987-4003,

https://doi.org/10.5194/amt-11-3987-2018, 2018.

---

## Author Comment (AC1) · 8 Dec 2018

We thank the reviewers for carefully reading our manuscript and for providing the critical review to improve the manuscript. In the following, we provide responses to both of the reviewers comments and concerns.

**Reviewer 1 comments:**
**Comment:** While the timeline link between molecular iodine release at low tide, iodine oxide formation (in daytime) and particle formation is well established (references cited in the paper itself), the study links these two with the formation of coastal mist

for the first time. I would then recommend a slightly more nuanced statement about the novelty of this 3-step time-line in the abstract and conclusions. What is certainly new and exciting about the study is the field observation of $I_2$ adsorbed onto water droplets. This appears to be a previously unknown stabilizing transport mechanism for the dispersal of $I_2$ in the marine environment. This finding is supported by targeted laboratory experiments demonstrating that the non-biological WIBS signals observed in the field are in fact consistent with the adsorption of $I_2$ on the surface of nebulized pure water droplets. **Manuscript changes:** Abstract and conclusion has been modified to reflect the reviewer comments.

*Page 2, Line 17:* While the process of molecular iodine release, particle formation and sea-fog formation have been studied in detail in previous studies, this study provides a potential link of the three phenomena.

*Page 28, Line 4:* The dual field and laboratory study presented here provides a possible real-time profile as observed on several occasions in July 2011 at Haulbowline Island, Co. Cork.

**Comment:** The absence of fluorescence for nebulized seawater in the presence of iodine vapour may well be explained by the formation of trihalide ions as the authors suggest. It would be interesting to know, though, whether the saline solutions prepared for the lab experiments have similar iodide and chloride concentrations to those expected in sea spray, and whether more diluted concentrations would have resulted in increased signal in the FL1 channel. Also, whether changing the residence time of the nebulized sea salt solution in the chamber could have shown some evidence of the kinetics of the removal of the adsorbed $I_2$ molecules at the surface. I any case, the characterisation of the sea salt solution should be included in section 2.3. I would like to see also some information about the pressure, temperature, flow rate, and residence time conditions in the aerosol dispersion chamber. **Response:** This study can indeed be expanded to flow tube experiments where there is a better control of the residence time. In terms of the experiment conditions, there was no pressure

gauge present in the system however the system can be considered to operate under atmospheric pressure. The system was allowed run for 30 minutes before and after each experimental run each experiment using the WIBS4 pump at 2.5 L/min. For all experiments (the iodine vapour with water droplets, sea salt droplets and mixed iodine/water droplets), the air flow in the system was 5.6 L/min for generating the aerosolised water droplets. For the solely iodine vapour experiments, only sublimed iodine was released into the system (No carrier gas), again after the system was pumped down for 30 minutes after each run. With regards to the residence time, there was no adjustable injector to vary reaction distance in the chamber. The only possibility was changing the direction of the injection (pointing the water aerosoliser upwards allowed for more adequate mixing time). **Manuscript changes:** *Page 8, Line 1:* 0.25g of $\geq$ 99% iodine crystals were measured for each sublimation test, with 0.05 g of refined rock salt from the Wieliczka Salt Mine used for the saltwater mimic tests. A smaller quantity of salt was used to avoid overloading the detector.

*Page 8, Line 24:* Before each experiment, the system was pumped down for 30 minutes to remove any residual material using the WIBS-4A total pumping capacity of 2.5 L/min. During the experiment, a flow of 5.6 L/min of compressed air was supplied for the aerosolization into the system. No pressure transducer was present in the system at the time so estimated conditions were of the order of 1 atm at 298 K. Relative humidity was >70 % based from observation of the chamber rather than direct measurement.

*Page 9, Line 7:* This study could be later applied at a flow tube experiment where there is a greater control of experiment conditions such as residence time.

*Page 9 Line 11:* The WIBS fluorescence data obtained in the experiments were filtered using thresholds most commonly utilized in the literature (ie the mean of forced trigger mode values + $3\sigma$ method) (Hernandez et al., 2016):

**Comment:** The absence of fluorescence when only iodine vapor is ad-mitted in the laboratory chamber is very interesting, but I don't find very

convincing the argument given by the authors to explain this point. The WIBS-4 instrument is optimized for particle detection, but in its configuration (http://www.dropletmeasurement.com/widebandintegrated-bioaerosol-sensor-wibs-neo) I don't find any obstacle for the detection of gas-phase fluorescence (perhaps the authors could comment more on that to make it clearer). $I_2$ molecular iodine presents a strong absorption feature in the 170-210 nm spectral range (the Cordes bands, D-X system), with peak absorption cross section of 2E-17 $cm^2$ molecule$^{-1}$ at 188 nm (Myer and Samson, 1970; Roxlo and Mandl, 1980). After absorption of VUV and middle UV photons, fluorescence from the D ion-pair state back to the ground state exhibits an ordinary bound to bound spectrum together with a bound to free diffuse quantum interference spectrum (the McLennan bands) (Tellinghuisen, 1974; Exton and Balla, 2004). Concurrently, a significant fraction of the initial D state population is collisionally transferred to the D' state at increasing buffer gas pressure, resulting in fluorescence in the D'-A' band at 340 nm. Since the transition probability of the D-X system at 280 nm is small (absorption cross section of 6E-19 $cm^2$ molecule$^{-1}$ (Saiz-Lopez et al., 2004)), this may explain why $I_2$ is not observed in the gas phase. On the other hand, complexation with water may red-shift the absorption spectrum. E.g. the peak of the $I_2$ VUV band shifts to 203 nm in aqueous solution (Kireev and Shnyrev 2015). This would be a plausible explanation as to why 280nm-pumped fluorescence in the 310 - 400 nm range can be observed when $I_2$ is complexed with water and not in the absence of water droplets. In feel that some spectroscopic discussion in this sense is pertinent. **Response:** Work indicated by Alizadeh et al., 2012 showed an $I_2$ peak at 450 nm but also a trace absorption form 250-290 nm (See figure 1 of that paper). O'Driscoll et al., 2008, discusses $I_3^-$ at 288 nm and 352 nm (both broad enough to meet the 280 nm and 370 nm flash lamp requirements of the WIBS). During the UV absorption analysis, the 288 nm and 352 nm peaks were observed, indicative of $I_3^-$. However, the 450 nm $I_2$ peak was also present in the spectra, indicative of some equilibrium between the two. The aerosolized mixture of iodine and water should have given a fluorescent signal if the $I_3^-$ ion was fluorescent as it was present in UV-absorption analysis but this

wasn't the case. The likely explanation for the observed FL1 fluorescence is the $I_2$ molecule adsorping to the water droplet surface. Work by (Liu et al., 2004) show that if $I_2$ is complexed with different organic solvents such as toluene, a strong absorption band is present at 280 nm due to the $I_2$ molecule red-shifting with the organic complex. A similar process is possible, with the $I_2$ not residing in gaseous or liquid phase but rather binded to the surface of the water droplet. The reason for no iodine vapour fluorescence is simply due to the vapour not existing in the particle phase for detection by the laser. If the laser doesn't detect a particle, then the flash lamps are not triggered, hence no fluorescence because no excitation. It's not stating that iodine vapour doesn't fluoresce, but that whatever the WIBS saw at Haulbowline wasn't attributed to that. One possibility is a non-fluorescent particle triggering the PMT with the laser, followed by triggering of the flash lamps, picking up $I_2$ fluorescence from surrounding vapour phase. However, after pumping the chamber down for 30 minutes, no particles are detectable, ensuring sufficient vacuum. **Manuscript changes:** Iodine vapour will be removed from the table as it may cause confusion in relation to the particle measurements.

*Page 21, line 19:* Iodine vapour fluorescence was not measurable because it does not provide particles which are necessary for detection by the internal diode laser of the WIBS. If the laser does not detect a particle of the specified size, the flash lamps are not triggered and therefore fluorescence cannot be detected because no excitation occurs. One possibility is that a non-fluorescent particle could be detected, which activates the flash lamps and a fluorescent signal from the surrounding iodine vapour is detected. However, with the chamber being pumped down for 30 minutes before each experiment, no particles of that size can interfere.

*Page 23, line 21:* The $I_3^-$ peak at 352 nm was not present at all in cold saltwater mimic samples and only in trace amounts (<5% of iodine and water mix) in heated samples while the peak at 288 nm largely remained. Work indicated by Alizadeh et al (2012) showed an $I_2$ peak at 450 nm but also a trace absorption at 250-290 nm. Therefore, it shows the possibility of $I_2$ absorbance at 280 nm even though the expected cross

section of $I_2$ at 288 nm is <1 x 1018 cm$^2$ (Roughly 18 times smaller than the peak absorbance at 203 nm (Saiz-Lopez et al., 2004). The absorbance spectra for each sample is available in the supplementary material.

*Page 24, line 19:* It is known for $I_2$ in vapour form that the transition probability of the D-X system at 280 nm is <1 x 1018 cm$^2$. This value is about eighteen times smaller than the peak absorbance at 203 nm (Saiz-Lopez et al., 2004). However work by (Liu et al., 2004) show that if $I_2$ is complexed with different organic solvents such as toluene, a strong absorption band is present at 280 nm due to the molecular $I_2$ UV spectrum red-shifting. A similar interaction can be envisioned for iodine bound to the surface of water droplets rather than being present in the vapour phase or simply dissolved inside a droplet.

**Comment:** Section 2.3 mentions that fluorescence spectra of the solutions were investigated using a Shimadzu RF-6000, but the results of these investigations are not reported. I also find that the solution absorption data is presented in a rather schematic way and that it would be more informative to show absorbance spectra (perhaps in the supplementary material) to demonstrate how efficient is the absorption of the two WIBS wavelengths. **Response:** The main implications of the spectra: Iodine and Milli-Q water – Gives $I_2$ at 288 nm and 450 nm as well the tri-iodide $I_3^-$ at 288 nm and 352 nm. Iodine and saltwater mimic – Initially only iodine chlorides present. Later waiting periods show retention of the 288 nm and 450 nm peak with the 352 nm reduced. **Figures:** Figure 1: UV Absorption spectrum of iodine in milliQ water with a range from 250 – 600 nm, Figure 2: UV Absorption spectrum of iodine in saltwater mimic, Figure 3: UV Absorption spectrum of iodine in a saltwater mimic after 4 days

**Manuscript changes:** The spectra has been added to the supplementary data and reference to the table in the paper.

**Comment:** The suggested link between $I_2 \cdot (H_2O)x$ and $HIO_3$ is a problematic one. Sipila et al. 2016 did observe $HIO_3$ and molecular cluster formation in their laboratory experiments, but in the presence of water vapor (no nebulized water droplets). Daly et al. do not report WIBS measurements of a mixture of water and iodine vapor, but it is known that $I_2$ and $H_2O$ form a weakly bound complex (Galvez et al. 2013), and under atmospheric conditions only a residual amount of $I_2$ would be complexed with $H_2O$. The presence of $I_2 \cdot (H_2O)x$ in the laboratory experiments of Sipila et al. is therefore unlikely. Furthermore, $HIO_3$ increase was observed by Sipila et al. in the field at daytime, well after noon, while Daly et al show that $I_2 \cdot (H_2O)x$ is a night time reservoir which disappear quickly after sunrise. **Response:** The work by Galvez et al., 2013 suggests a theoretical weakly bound complex of one $I_2$ molecule with one $H_2O$ molecule (14 kJ mol$^{-1}$). Further studies should address apply the iodine/water vapour method to check if this is a possibility. However, a different mechanism is suggested here. Several $I_2$ molecules would bind to one $H_2O$ droplet as a surface ionic mechanism. The interaction between iodine and oxygen from the water droplet may result in enough partial positivity in the iodine atom to make it susceptible to radical attack, thus starting the reaction process to $HIO_3$. In the paper, the WIBS-4 peak for fluorescent counts at 6 am was during the middle of July (approaching the longest day of the year where sunrise would be before 6 am) whereas the observations by Sipila et al., 2016 were made from August to October. At this point, morning daylight would have started later sometime between 6:30-8:30 am. There is 3 hours of delay between the WIBS signal decrease and the SMPS count increase. If the $HIO_3$ and IO data (Figure 1.) was recorded towards the October period, the later sunrise could correlate the observed $HIO_3$/IO traces to the study here. **Manuscript changes:** *Page 26, line 22:* Work by Galvez et al (2013) suggests a theoretical weakly bound complex of one $I_2$ molecule with one $H_2O$ molecule (14 kJ mol$^{-1}$). However, the current study likely addresses cases when several $I_2$ molecules bind to one or more water droplets in a surface adsorption mechanism.
*Page 27, line 7:* In fact, a recent report, which outlines evidence for some coastal

aerosol particle formation being due, in part, to the sequential addition of $HIO_3$ indicates that at Mace Head, Ireland, the production of the oxo-acid has been shown to begin at sunrise reaching a maximum at noon (Sipila et al., 2016). It should be noted that the measurements were made during the August to October period. During that time, the iodic acid is reported to appear at 13.00 with the IO radicals peaking at 15.00 pm due to later sunrise.

**Minor Comments:**

Comment: Page 6, line 20: 0.5-20 m. do you mean micron?
Response: Yes this was meant to be micron.
Paper edit: This change has been included in the paper.

Comment: Page 11, lines 2 and 3: it looks like all these size ranges should be microns rather than meters.
Response: Yes they should be.
Paper edit: This change has been included in the paper.

Comment: All figures. In general, the legends and axis labels are too small and difficult to read, especially in multi-panel figures.
Paper Edit: Each graph has been re-edited to include larger legends and axis labels.

Comment: Figure 9: some of the tidal height data is missing: the 6:00AM and 18:00 tidal values are not shown (as opposed to figure 1 in the supplementary information).
Paper edit: This change has been included in the paper.

**Reviewer 2 comments:**

**Comment:** I think Section 2 needs a more detail on A) the laboratory experiments that took place using the WIBS-4A instrument and B) the preparation for the data for both the field analysis and complementary laboratory studies. **Response:** The manuscript has been modified to reflect these comments. **Manuscript changes:** As already indicated in response to reviewer 1

**Comment:** From my understanding the author uses the FT signals as the fluorescence threshold, and compares results to what was seen in the Hernandez et al., 2016 publication, however this publication uses the default FT + 3 $\sigma$ threshold. **Response:** The paper has been rewritten to correct any ambiguities within regard to the threshold used for the work. The default mean Forced Trigger + 3 $\sigma$ threshold was used **manuscript changes**: *Page 9 Line 11:* The WIBS fluorescence data obtained in the experiments were filtered using thresholds most commonly utilized in the literature (ie the mean of forced trigger mode values + 3$\sigma$ method) (Hernandez et al., 2016):

**Comment:** laboratory and field study should be more explicit (e.g. size calibration information, fluorescence calibration information, the fluorescence threshold chosen-whether it is the average, median, etc.). Size calibrations were carried out using several PSL sphere ranges 0.5, 0.82, 1, 2, 4, 10, 12 microns **Response:** At the time of the work the Robinson fluorescence calibration paper had not been published. Thus a fluorescence calibration was not under taken. **Manuscript changes:** *Page 9, Line 23:* For the WIBS-4A instrument, size calibrations were carried out using Polystyrene Latex Spheres (PSL) with diameters 0.5, 0.82, 1, 2, 4, 10, 12 $\mu$m. The internal photomultipliers for each WIBS were not measured at the time.

**Comment:** Section 2.2 Field Instrumentation: On page 6 lines 21-24, the author states that both the WIBS-4 and the WIBS-4A units were identical in terms of functionally-this is strong statement. **Response:**We agree with the reviewer and have modified

this point significantly. **Manuscript changes:** The section has been edited to read *Page 6, Line 31:* Both instruments display similarities in terms of sampling methods and build but have a few distinctions such as the WIBS-4 dual gain detection approach and the WIBS-4A double threshold system. The WIBS-4A has a slightly higher flow rate at 2.5 L/min and 300 ml/min (flow velocity of 18 m/s) compared to the WIBS-4 at 2.4 L/min and 230 ml/min (flow velocity of 12 m/s). Similarly, variation in fluorescent intensity between WIBS instruments for identical particles is a potential problem in such studies a problem which has been discussed previously in the literature. (Robinson et al., 2017, Savage et al., 2017, and Könemann et al., 2018)

**Comment:** Can the author please comment on whether such calibration (fluorescence) was done? Where the PMT voltages measured for each WIBS unit? **Response:** No as explained in response to reviewer 1, the PMT voltages were not measured

**Comment:** Page 11, lines 5- 20: In general, I think this section needs more discussion in regards to the suggested publications and their analysis strategies - Gabey et al., 2010, Perring et al., 2015, Savage et al., 2017 and Savage et al., 2018. It is not clear what the author means by stating " Unusually, fluorescence signals were mainly measurable in the FL1 channel. FL2 registered little emission above threshold as illustrated in Figure 4, which shows plots of size/AF data as a function of the FL1 and FL2 channels. (FL3 showed no fluorescence). The larger size feature (2-6 um) consisting of highly fluorescent solid particles/droplets but only in the FL1 channel represents a behaviour that has not been observed previously in any WIBS field campaign. Hence fungal spores, certain pollen and bacteria as large as 2 um (Hernandez et al., 2016) can be found in the 2-6 um size regime but are fluorescent in all channels because of their amino acid, tryptophan and NAD(P)H contents". **Response:** We agree with the reviewer and the section has been rewritten to reflect this as follows. **Manuscript**

**changes:**   *Page 11, Line 19:* The work by Hernandez et al (2016) suggests that some fungal spores show fluorescent characteristics that are present in FL1 but not FL3. However, the conditions on site would not favour spore release as the island has very little soil-based vegetation with only sea kelp present. Very low wind speeds were recorded during the measurement periods (<2.5 m/s during the 15th – 16th July and <5 m/s during the 26th-28th July). Therefore, it was highly unlikely that material could be carried on to the island from the mainland. In any case, the particles are less likely to be bacteria or pollen because of size constraints. Hence bacteria sizes are found at the lower limit (and below) of WIBS detection. Pollen sizes are often measured at much higher than the upper limit of WIBS detection and so are generally captured by the particle trap. During the summer period, the dominant fungal spore in locations close to but not at the Cork Harbour coastline, is known to be Cladosporium which is generally released during the day time (10:00 am - 12:00 pm onwards) and under dry conditions (O'Connor et al., 2015, Healy et al., 2014). By contrast in this study WIBS particle detection was found in the night-time period between 00:00 – 08:00 am.
*Page 12, line 11:* Ascospores are linked with rain releases but only 0.2 mm of rainfall was recorded after 09:00 am on the 16th July, after which the WIBS signal is seen to decease (O'Connor et al., 2015).

**Comment:** Several studies suggest there are non-biological, fluorescent particles that may be interferences when discriminating between bio vs. non-biological particles, and even different particle types (Huffman et al., 2010, Pohlker et al., 2015, and Savage et al., 2017, and references there-in). Can the author comment on these possible interferences, and if these substances were taken into consideration during their field analysis? **Manuscript changes:**   The manuscript has been updated to reflect the comments of the reviewer. *Page 20, Line 12:* It should be not that other non-biological particles have been seen to be fluorescent. Mineral dust was also considered as a potential source of fluorescent particles. In fact, studies have shown that fluorescent mineral dust can contribute up to  10% of the total measured (Toprak and Schnaiter.,

2013). However in this study, using the FL1 and FL3 channel filters this dust artefact was removed. A lack of FL3 fluorescence signals in this study rules out the presence of mineral dust because it weakly fluoresces in the FL1 and FL3 detection ranges and therefore is considerably weaker than biofluorophore signals (Toprak and Schnaiter, 2013; Pohlker et al., 2012). Polycyclic Aromatic Hydrocarbons (PAH's) could also be considered a potential interference to the measurements made here due to their highly fluorescent nature but since they are largely present on the surface of soot particles which generally exist in submicron sizes, detection by the WIBS is unlikely unless oil droplets were present. The complex chemical environments associated with soot particles can also lead to fluorescence quenching of PAH's (Pohlker et al., 2012). Humic-like substances (HULIS) and secondary organic aerosols (SOA) have also been indicated as potential interference signals in the literature (Pohlker et al., 2012).

**References added:**
Healy, D., Huffman, J., O'Connor, D., Pöhlker, C., Pöschl, U. & Sodeau, J. 2014. Ambient measurements of biological aerosol particles near Killarney, Ireland: a comparison between real-time fluorescence and microscopy techniques. Atmospheric Chemistry and Physics, 14, 8055-8069.

Liu, Z., Tian, J., Zang, W., Zhou, W., Song, F., Zhang, C., Zheng, J., and Hua Xu, (2004) Flexible alteration of optical nonlinearities of iodine charge-transfer complexes in solutions, Opt. Lett. 29, 1099-1101

O'connor, D., Healy, D. & Sodeau, J. 2015. A 1-month online monitoring campaign of ambient fungal spore concentrations in the harbour region of Cork, Ireland. Aerobiologia, 1-20.

Pöhlker, C., Huffman, J. & Poschl, U. 2012. Autofluorescence of atmospheric

bioaerosols-fluorescent biomolecules and potential interferences. Atmos. Meas. Tech, 5, 37-71.

Toprak, E. & Schnaiter, M. 2013. Fluorescent biological aerosol particles measured with the Waveband Integrated Bioaerosol Sensor WIBS-4: laboratory tests combined with a one year field study. Atmospheric Chemistry and Physics, 13, 225.

Robinson, E. S., Gao, R.-S., Schwarz, J. P., Fahey, D. W., and Perring, A. E.: Fluorescence calibration method for single-particle aerosol fluorescence instruments, Atmos. Meas. Tech., 10, 1755– 1768, https://doi.org/10.5194/amt-10-1755-2017

Savage, N. J., Krentz, C. E., Könemann, T., Han, T. T., Mainelis, G., Pöhlker, C., and Huffman, J. A.: Systematic characterization and fluorescence threshold strategies for the wideband integrated bioaerosol sensor (WIBS) using size-resolved biological and interfering particles, Atmos. Meas. Tech., 10, 4279-4302, https://doi.org/10.5194/amt10-4279-2017, 2017.

Savage, N. J. and Huffman, J. A.: Evaluation of a hierarchical agglomerative clustering method applied to WIBS laboratory data for improved discrimination of biological particles by comparing data preparation techniques, Atmos. Meas. Tech., 11, 4929-4942, https://doi.org/10.5194/amt-11-4929-2018, 2018.

Könemann, T., Savage, N. J., Huffman, J. A., and Pöhlker, C.: Characterization of steady-state fluorescence properties of polystyrene latex spheres using off- and online spectroscopic methods, Atmos. Meas. Tech., 11, 3987-4003,

ACPD

Interactive
comment

[Figure]

**Fig. 1.** UV Absorption spectrum of iodine in milliQ water with a range from 250 – 600 nm

[Figure]

**Fig. 2.** UV Absorption spectrum of iodine in saltwater mimic

[Figure]

**Fig. 3.** UV Absorption spectrum of iodine in a saltwater mimic after 4 days

---

## Referee Report (RR1)

I'm generally satisfied with the authors responses to my comments. However, there are still a couple of points that need clarification before the paper is published:

p 7, section 2.3 (laboratory experiments). In my previous report I asked the authors to characterize the sea salt solution employed in the lab experiments to produce sea-spray mimics. This request has been mostly ignored. I think this information is necessary because of the explanation given by the authors about the absence of fluorescence from nebulized 'seawater' in the presence of iodine vapor in the lab (the formation of tri-halide ions at the surface). Sea-spray surely has a given Cl- concentration - why is then the quenching effect of trihalides not happening in the field? I suspect this might be because the Cl- concentration in the lab mimics is much higher than in real sea-spray.

p 21, line 21. Do you mean NO fluorescence is detected?

p. 27, lines 6-12. HIO3 is said to appear at sunrise in p.9, and then to appear at 13: 00 in line 11. I cannot find any information about IO radicals peaking at 15:00 pm anywhere in Sipila's paper - what they plot in figure 1 alongside HIO3 are iodine oxide clusters.

---

## Author Response (AR2)

Dear editor,

We thank you for the feedback. Please see below for our response and revised manuscript with marked changes following the minor revisions.

**Comment:** p 7, section 2.3 (laboratory experiments): Characterize the sea salt solution.
**Manuscript edit:** The following was added to the manuscript:

The following composition was present in 35 ppt of saltwater solution (3.5 g of the rock salt in 100 ml of water):
Cl-      - 55.29%
Na+     - 30.74%
Mg2+   - 3.69%
SO42-  - 7.75%
Ca2+    - 1.18%
K+       - 1.14%
The Cl- and Na+ ions make up 86% of the total Salinity.

**Comment:** p. 21, line 21: Do you mean NO fluorescence is detected?
**Answer:** The sentence was meant to state that no fluorescence is detected because no particle in the WIBS size range is detected by the laser.
**Manuscript edit:** If the laser does not detect a particle of the specified size, the flash lamps are not triggered and therefore **no** fluorescence is detected because no excitation occurs.

**Comment:** p.27, lines 6-12: HIO3 is said to appear at sunrise in p.9 and then to appear at 13:00 in line 13. I cannot find any information about IO radicals at 15:00pm anywhere in Sipila's - what they plot in figure 1 alongside HIO3 are iodine oxide clusters.
**Answer:** The reviewer is correct in saying that iodine oxide clusters and not IO were stated in the Sipila et al., 2016 paper. However it is highly likely IO radicals are present just before these clusters form.

[revised manuscript text omitted]